# Landscape of multi-nucleotide variants in 125,748 human exomes and 15,708 genomes

Qingbo Wang [1,2,3], Emma Pierce-Hoffman[1], Beryl B. Cummings[1,2,4], Jessica Alföldi [1,2], Laurent C. Francioli [1,2], Laura D. Gauthier[1,5], Andrew J. Hill[1,6], Anne H. O'Donnell-Luria [1,2], Genome Aggregation Database Production Team, Genome Aggregation Database Consortium, Konrad J. Karczewski [1,2] & Daniel G. MacArthur [1,2,7,8]*

Multi-nucleotide variants (MNVs), defined as two or more nearby variants existing on the same haplotype in an individual, are a clinically and biologically important class of genetic variation. However, existing tools typically do not accurately classify MNVs, and understanding of their mutational origins remains limited. Here, we systematically survey MNVs in 125,748 whole exomes and 15,708 whole genomes from the Genome Aggregation Database (gnomAD). We identify 1,792,248 MNVs across the genome with constituent variants falling within 2 bp distance of one another, including 18,756 variants with a novel combined effect on protein sequence. Finally, we estimate the relative impact of known mutational mechanisms - CpG deamination, replication error by polymerase zeta, and polymerase slippage at repeat junctions - on the generation of MNVs. Our results demonstrate the value of haplotype-aware variant annotation, and refine our understanding of genome-wide mutational mechanisms of MNVs.

[1] Program in Medical and Population Genetics, The Broad Institute of MIT and Harvard, Cambridge, MA 02142, USA. [2] Analytic and Translational Genetics Unit, Massachusetts General Hospital, Boston, MA 02114, USA. [3] Program in Bioinformatics and Integrative Genomics, Harvard Medical School, Boston, MA 02115, USA. [4] Program in Biomedical and Biological Sciences, Harvard Medical School, Boston, MA 02115, USA. [5] Data Sciences Platform, Broad Institute of MIT and Harvard, Cambridge, MA 02142, USA. [6] Department of Genome Sciences, University of Washington, Seattle, WA 98195, USA. [7] Centre for Population Genomics, Garvan Institute of Medical Research, and UNSW Sydney, Sydney, Australia. [8] Centre for Population Genomics, Murdoch Children's Research Institute, Melbourne, Australia. A full list of consortium members appears at the end of the paper. *email: danmac@broadinstitute.org

Multi-nucleotide variants (MNVs) are defined as clusters of two or more nearby variants existing on the same haplotype in an individual[1,2] (Fig. 1a). When variants in an MNV are found within the same codon, the overall impact may differ from the functional consequences of the individual variants[3]. For instance, the two variants depicted in Fig. 1b are each predicted individually to have missense consequences, but in combination result in a nonsense variant. Such cases, which would be missed by virtually all existing tools for clinical variant annotation, can result both in missed diagnoses and false positive pathogenic candidates in analyses of families affected by genetic diseases[1,2].

MNV identification tools[4–8] have been applied to databases of human genetic variation at varying scales, including 1000 Genomes[9] Phase 3 (2504 individuals with high coverage exome and low coverage genome-sequencing data), and the Exome Aggregation Consortium[1] (60,706 individuals with high coverage exome data). Together, these analyses identified over 10,000 MNVs altering protein sequences, demonstrating the pervasive nature of MNV annotation in the population-level data. In addition, analysis of the 1000 Genomes data set highlighted differences in the frequencies of MNVs depending on sequence context[10]. In combination with yeast experiments[11–13], biological mechanisms that account for the enrichment of specific types of MNVs, such as DNA replication error by polymerase zeta, have been suggested.

Studies of newly occurring (de novo) MNVs have also been performed using trio data sets[2,14–16]; analysis of 283 trios with whole-genome sequence data[16] confirmed that MNV events occur much more frequently than expected by random chance. By

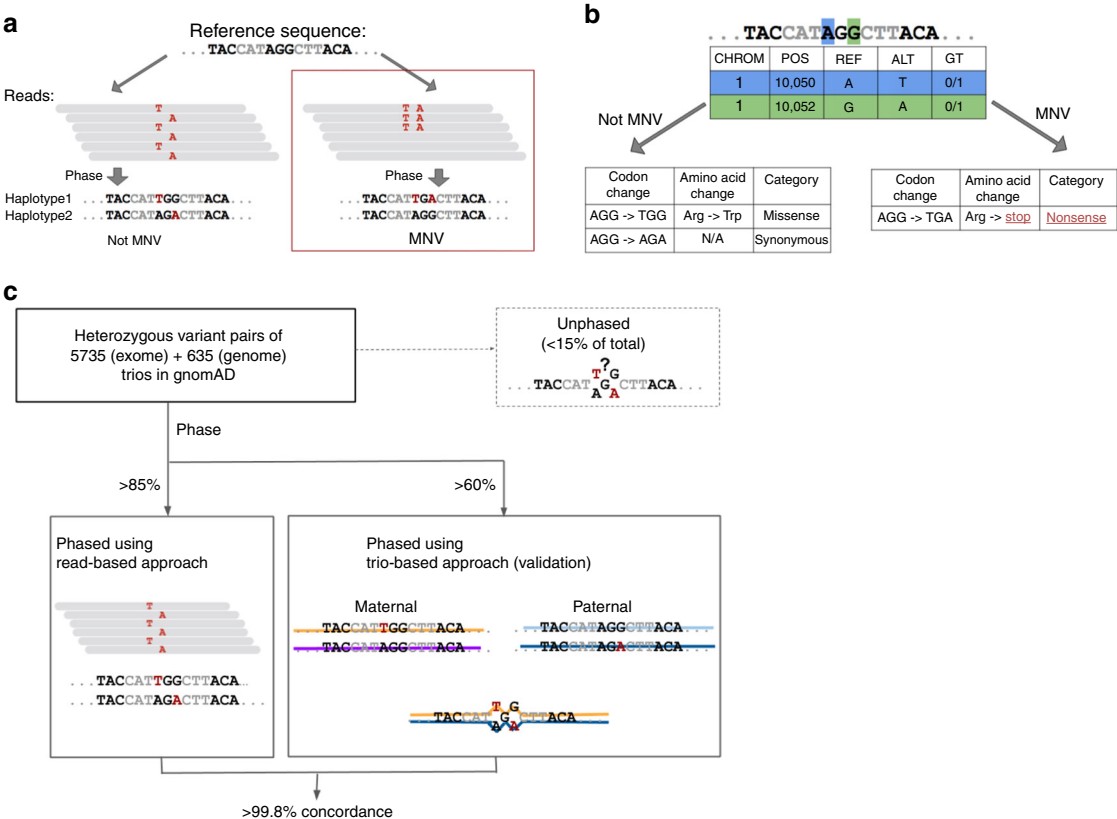

**Fig. 1** Definition and an example of MNVs, and validation of phasing sensitivity. **a** Definition and an example of an MNV. In this paper, an MNV is defined as two or more nearby variants existing on the same haplotype in the same individual. **b** Impact of MNVs in coding regions. The amino acid change caused by an MNV can be different from either of the individual single-nucleotide variants, which creates the potential for missannotation of the functional consequence of variants. **c** Graphical overview of the analysis of phasing sensitivity and specificity using trio samples from our gnomAD callset. We identified all heterozygous variant pairs that pass quality control (see the Methods section) and compared the phase information assigned by read-based phasing with that of trio-based phasing

focusing on noncoding regions, this study also highlighted potentially different mechanisms that dominate MNV generation depending on the genomic region and the distance between the two constitutive variants. As part of the Deciphering Developmental Disorders (DDD) study[17], Kaplanis et al.[2] analyzed exome-sequence data from over 6000 trios to quantify the pathogenic impact of MNVs in developmental disorders, showing that such variants are substantially more likely to be deleterious than SNVs and further clarifying the mutational mechanisms that generate them. These analyses also have provided estimates of the germline MNV rate per generation, falling into a consistent range of 1–3% of the SNV rate. Although these studies have provided valuable information about the mutational origins and functional impact of MNVs, to date there has been no analysis that investigated MNVs across the entire genome (including noncoding regions) in many thousands of deeply sequenced individuals, limiting our understanding of the genome-wide profile and complete frequency distribution of this class of variation.

Here, we present the analysis of a large-scale collection of MNVs, along with clinical interpretation of MNVs from over 6000 sequenced individuals from rare disease families. We also provide gene-level statistics on MNVs and describe the distribution of MNVs by functional consequence and by gene-level constraint. Finally, to enhance our understanding of MNV mechanisms, we examine the distributions of MNVs stratified by more than ten different functional annotations across the human genome, as well as estimates of the genome-wide per-base frequencies of the dominant mutational processes generating MNVs.

## Results

### Read-based phasing for identification of MNVs.
Identification of MNVs requires the constituent variants to be properly phased —that is, to be identified accurately as either both occurring on the same haplotype (in *cis*) or on two different haplotypes (in *trans*). Phasing can be performed following three broad strategies: read-based phasing[18], which assesses whether nearby variants co-segregate on the same reads in DNA sequencing data; family-based phasing[19], which assesses whether pairs of variants are co-inherited within families; and population-based phasing[20], which leverages haplotype sharing between members of a large geno-typed population to make a statistical inference of phase. Read-based phasing is particularly effective for pairs of nearby variants, making it suitable for the analysis of MNVs.

For this project, we generated read-based phasing results for variants in the Genome Aggregation Database (gnomAD) v2.1 callset using GATK HaplotypeCaller[21], yielding 125,748 human whole exomes and 15,708 genomes with local phase information; the properties of this callset are described in detail in an accompanying paper[22]. To assess phasing accuracy, we used 5785 family trios with exome-sequencing data and 635 family trios with whole-genome sequencing data that largely overlapped with the gnomAD 2.1 release data. We calculated the phasing sensitivity, defined as the fraction of heterozygous variant pairs that have read-based phase information assigned for both variants, and found that it was 87.9% for adjacent heterozygous variant pairs, reflecting the stringent haplotype-calling criteria of GATK[21] (Supplementary Tables 1–3). We used Phase-By-Transmission (PBT)[19], a family-based phasing method (Fig. 1c), to assess our phasing specificity, and found that over 99.8% of the MNVs identified with read-based phasing were consistent with the PBT trio-based phasing. The sensitivity and specificity of our read-based phasing remained high even when the two variants of the MNV were 10 bp apart (82.8% and 99.8%; Supplementary Fig. 1 and Supplementary Table 1). These results demonstrate

high specificity and sensitivity for the detection of MNV events across the genome.

### Functional impact of MNVs.
In order to provide an overview of the functional impact of MNVs (Fig. 1b), we examined all phased high-quality SNV pairs (i.e., SNV pairs that pass stringent filtering criteria; see the Methods section) within 2 bp distance of each other across the 125,748 exome-sequenced individuals from our gnomAD 2.1 data set, resulting in the discovery of 31,575 MNVs exist within the same codon. When the two variants comprising the MNV were considered together, the resulting functional impact on the protein differed from the independent impacts of the individual variants in ~60% of cases (18,756 MNVs; Fig. 2a; Supplementary Data 1). Among the differing annotations of functional consequence, 407 were gained nonsense (neither individual SNV was a nonsense mutation, but the resulting MNV is), and 1821 were rescued nonsense (at least one of the two individual SNVs would create a nonsense mutation, but the resulting MNV does not). Such categories of MNVs have a major impact on variant interpretation, and thus are critical for accurate variant annotation. There was an average of 55.2 variants with altered functional interpretation (including 0.062 gained and 4.42 rescued nonsense) due to MNVs per individual.

To understand the overall impact of correctly annotating the functional consequence of MNVs in a population-level data set, we counted the number of gained/rescued nonsense mutations per gene in gnomAD (Fig. 2b; Supplementary Data 1). For rescued nonsense mutations, we found 1538 sites that are rescued in all the individuals with the component variants. A total of 1633 genes carried gained or rescued nonsense mutations within our data set, including 41 genes that are disease-relevant (reported by OMIM[23] or annotated as haploinsufficient by Clingen[24,25]). In addition, the proportion of rescued nonsense mutations of falling in predicted loss-of-function (pLoF) constrained genes (genes with a significant depletion of pLoFs compared with an expectation based on a mutational model[1,26], defined as LOEUF[22] decile <20%) was higher (proportion = 0.219) when compared with all the other classes of MNVs (proportion = 0.192; Fisher's exact test, $p = 0.0247$; Fig. 2c; Supplementary Fig. 2). Conversely, gained nonsense mutations are depleted among constrained genes (proportion = 0.0620) compared with all other classes of MNVs (Fisher's exact test, $p = 1.01 \times 10^{-11}$). These results suggest a significant enrichment of LoF annotation errors in the absence of MNV annotation.

In addition, we have investigated another class of variant pairs whose combined interpretation can be highly different from either of the individual component variants: insertion/deletion (indel) pairs that result in frame restoration (e.g., 4 bp deletion + 7 bp insertion, resulting in 3 bp = 1 amino acid insertion), and have annotated such frame-restoring indel pairs ($n = 1406$) when separate by up to 30 bp (considering the limitations of read-based phasing; Supplementary Fig. 3). When we compare the LoF confidence of constituent indels, we found that the proportion of frame-restoring indel pairs falling on LoF-constrained genes were significantly higher when the constituent indels are high-confidence (HC) LoFs (proportion = 0.0262 for low-confidence, LC, and 0.167 for HC pairs. Fisher's exact test, $p = 1.66 \times 10^{-7}$; Supplementary Fig. 3h), suggesting that frame-restoring indel pairs can also be a source of LoF annotation errors.

Finally, in order to understand the impact of these variants in clinical applications, we also annotated MNVs in 6072 sequenced individuals from rare disease families, including 4275 case samples. This resulted in 16 gained nonsense mutations and 110 changed missense MNVs with high CADD[27] scores and low frequencies in gnomAD (CADD >20 and <10 individuals in

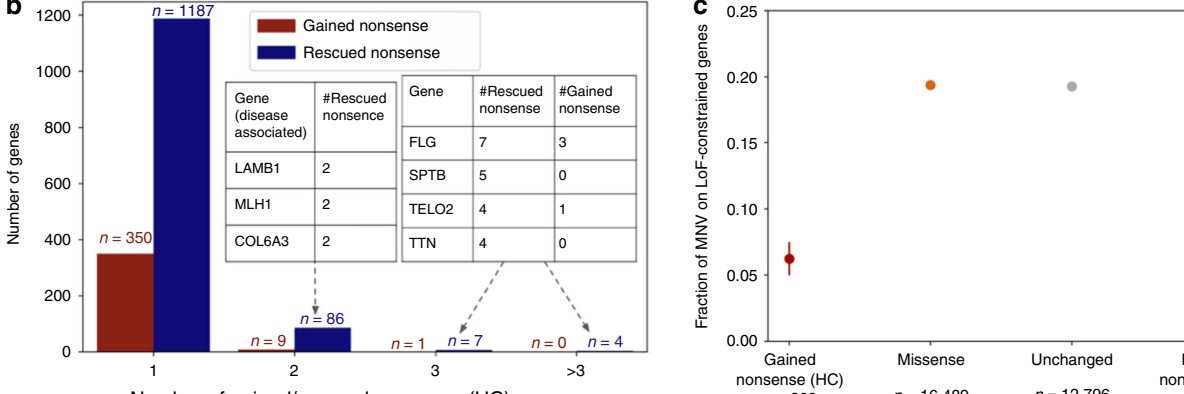

**Fig. 2** Functional impact of MNVs. **a** The number of MNVs in the gnomAD exome data set per MNV category. Of the 1821 rescued nonsense mutations, 1538 are rescued in all individuals that harbor the original nonsense mutation and are used for the analysis in (**b**) and (**c**). Gained and rescued nonsense MNVs were further filtered to HC pLoF in (**b**) and (**c**). **b** The number of gained/rescued nonsense mutations per gene, and examples of disease-associated genes with two or more gained/rescued nonsense mutations. **c** The fraction of each category of MNV found in a set of 3941 constrained genes (top two deciles of constraint[22])

gnomAD; Supplementary Data 2). However, after close manual curation, none of the corresponding MNVs were definitively causal variants for the diseases affecting the family, suggesting that MNVs contribute to only a small fraction of total rare disease diagnoses, in line with expectations based on their relative rarity and previous results[2].

**Genome-wide mutational mechanisms of MNVs**. We next turned our attention to understanding the mutational mechanisms underlying the origins of MNVs genome-wide, focusing on whole-genome sequence data from 15,708 individuals in the gnomAD v2.1 callset. We considered pairs of high-quality variants in autosomes separated by up to 10 bp, resulting in the assembly of a catalogue of 5,513,219 MNVs including 1,792,248 MNVs within 2 bp distance—an order-of-magnitude increase in size over previous collections.

We considered three established major categories of mutational origins of MNVs with constituent SNVs falling next to each other

(adjacent MNVs. Figure 3a), each of which is biased toward certain MNV patterns: (1) combinations of distinct single-nucleotide mutation events; (2) replication errors by error-prone polymerase zeta; and (3) polymerase slippage events at repeat junctions. MNVs in the first category are a product of two or more SNVs, which typically occur in different generations and may thus have different allele frequencies. We expect to see an enrichment of CpG transition compared with non-CpG transversion for this class, due to the underlying difference of SNV mutation rate[28–30]. The second category, replication error introduced by DNA polymerase zeta (pol-zeta), is a well known class of replication error that introduces MNVs. Previous studies[10–13,31] have shown that pol-zeta is prone to specific types of replication error, mainly TC- > AA, GC- > AA, and their reverse complements, with experimental evidence that these MNV patterns occur in a single generation; thus, the constituent SNVs will typically have the same allele frequencies. The third category, replication slippage, is another known mode of DNA replication error[32–34]. This process is especially frequent at sites

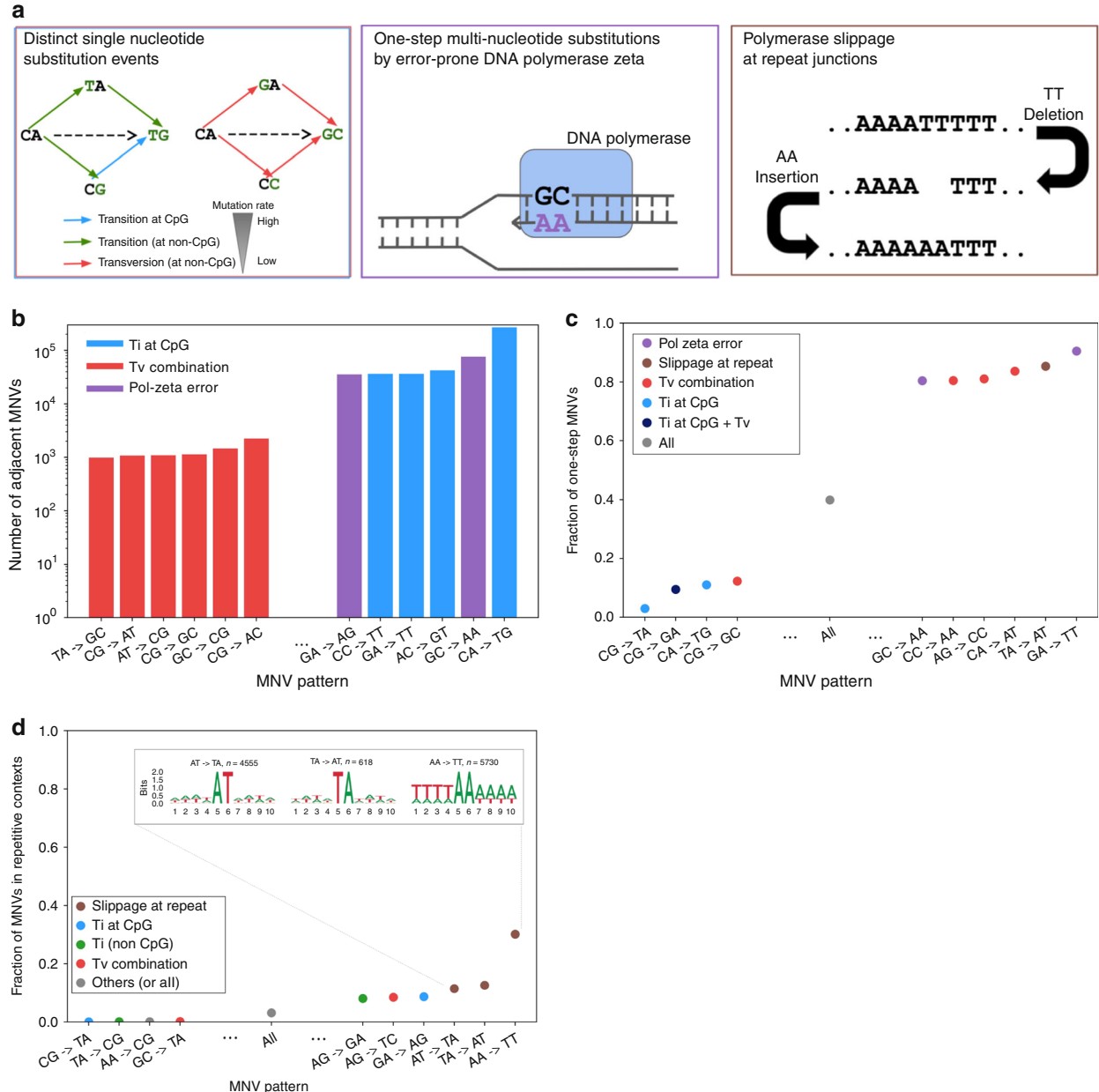

**Fig. 3** Mutational origins of MNVs. **a** Three major categories of the mutational origin of MNVs. (Left) A combination of single-nucleotide mutational events. Since the baseline global mutation rate is highly different between transversions and CpG and non-CpG transitions, even a simple combination of single-nucleotide mutational events could result in a highly skewed distribution of MNVs. (Center) One-step mutation caused by error-prone DNA polymerases. For this class of MNVs, since the two mutations occur at once during DNA replication, the allele frequency of the two constituent SNVs of the MNV is more likely to be equal. (Right) Polymerase slippage at repeat junctions. Mutation rates are highly elevated in repeat regions, and are therefore likely to cause various complex patterns of mutations, occasionally resulting in MNVs. **b** The log-scaled number of MNVs per substitution pattern. **c** The fraction of one-step MNVs per substitution pattern. Error bars represent standard error of the mean (often smaller than the dot size). **d** The fraction of MNVs that are in repetitive contexts, and bits representation[63] of sequence contexts. Error bars represent standard error of the mean. Colors in the bars in panels **b**–**d** represents the predicted major mechanism of MNVs for each substitution pattern

with repetitive sequence context; previous studies[35–37] have shown that the indel rate can be up to $10^6$ times higher than the SNV mutation rate at these sites. As shown in Fig. 3a, the combination of an insertion and then a deletion of two base pairs can result in an MNV.

We observed the signature of each of these MNV mechanisms in our data set. First, we calculated the number of MNVs for each MNV pattern (Fig. 3b) and observed that the most frequent MNV pattern is CA- > TG substitutions, which are likely to occur as a combination of an A- > G transition, followed by a high mutation rate C- > T CpG transition (Supplementary Fig. 4a). On the other hand, the least frequent MNV pattern is TA- > GC substitutions, which occur as a combination of two non-CpG transversions. The 273.4-fold difference (270,071 versus 988) of the frequency of MNVs between these two patterns is comparable with the theoretical ratio calculated based on the mutation rate of the component SNVs (475.6-fold), and the overall correlation between the theoretical and observed frequency of each MNV pattern was strong (Pearson correlation $r = 0.839$ with $p = 9.15 \times 10^{-22}$ in log space; Supplementary Fig. 4b–e).

To investigate the extent of pol-zeta signature, we calculated the number of MNVs in which the gnomAD allele counts of the constitutive single-nucleotide variants are equal (following previous methodology[2], also described in the Methods section), and observed that these one-step MNVs are significantly enriched in MNV patterns matching the pol-zeta signature (90.5% for GA->TT, and 80.5% for GC->AA, compared with 39.9% overall; Fisher's exact test, $p < 10^{-100}$; Fig. 3c).

Finally, in order to capture polymerase slippage events, we calculated the fraction of MNVs in repetitive contexts per MNV pattern (Fig. 3d). For the MNV patterns AA->TT, >30% of all the MNVs observed were in repetitive contexts. The fractions of the MNV patterns AT->TA and TA->AT in repetitive contexts were also high, exceeding 10% (Fisher's exact test, $p < 10^{-100}$ compared with the 3.15% across all patterns). For all MNV patterns in repeat contexts, we see a significant excess of MNVs compared with the expected number based on a model that assumes MNVs are simple combination of two SNV events (Supplementary Fig. 4). These observations support the role of replication slippage as one of the major drivers of MNVs. In addition, we did not see a correlation between the frequency of one-step MNVs and the frequency of MNVs in repetitive contexts (Pearson correlation $r = 0.0561$, $p > 0.05$; The fraction of one-step MNVs exceeded 80% for AT->TA and TA->AT, but was 46% for AA->TT), suggesting that multiple slippage events leading to MNV generation can take place either as a single event (i.e., in single generation) or multiple events (i.e., in different generation), or even recurrently. These findings come with the caveat that variants in repetitive regions will have higher error rates due to slippage and misalignment errors, but we have reduced this risk by applying random forest filtering for individual sites, as well as removing all the variants in low-complexity regions from our analysis (see the Methods section).

**Estimation of global mutation rate of MNVs.** In order to compare the frequency of three different mechanisms, we quantified the contribution of two single-nucleotide variation events vs other replication error modes, such as pol-zeta errors or replication slippage, using a simple probabilistic model. Specifically, focusing on adjacent MNVs, we assigned the MNV frequency for each MNV pattern to be the sum of the probability of two SNV events ($P$) and the probability of other replication error factors ($Q$), and estimated the $Q$ term. In other words, we estimated the divergence of the observed number of MNV sites from the number expected by a simple SNV mutation model (see the Methods section). The resulting estimated proportion of two SNV events and other replication error events is described in Fig. 4a.

As expected, the proportion differs substantially from one MNV pattern to another. For example, while 98.0% of CA->TG MNVs appear to be caused by combinations of simple SNV events, the corresponding proportion is 5.84% for GA->TT, 18.9% for GC->AA, and 9.52% for AA->TT MNVs. We presume that the lower proportion of two simple SNV events is mainly due to pol-zeta errors for GA->TT and GC->AA, and polymerase slippage for the AA->TT. Since 83.2% of the overall MNVs were classified as either SNV combination, repeat context, or pol-zeta error at GA->TT or GC->AA, our analysis suggests that these three major categories explain a substantial fraction of MNV events genome wide, although some possible additional mechanisms with smaller frequencies might exist. These calculations also allow us to estimate the genome-wide mutation rate of MNVs caused by pol-zeta: $1.59 \times 10^{-10}$ per 2 bp per generation for GA->TT, and $4.08 \times 10^{-10}$ for GC->AA. Given that there are ~$1.66 \times 10^8$ GA pairs and $1.20 \times 10^8$ GC pairs in the reference human genome, we estimate there are on average 0.026 GA->TT and 0.049 GC->AA mutations per generation (Supplementary Data 3).

We also explored the potential mutational mechanisms for MNVs with a greater distance between the component variants (Supplementary Figs. 5–7), and observed signatures of non-independence of mutation events extending over distances up to 10 bp, with an enrichment of motifs consistent with pol-zeta and polymerase slippage mechanisms for adjacent MNVs (minimum 1.08, maximum 4.06-fold enrichment of one-step MNV, Fisher's exact test, $p$-value < 0.05; Supplementary Figs. 8,9). This confirms the presence of mutational mechanisms capable of creating simultaneous mutations separated by considerable distances[16,29,38–40], although further work will be required to fully characterize the underlying processes.

Overall, our analysis of MNVs in 15,708 whole-genome-sequenced individuals supports the previously suggested three major mechanism of MNVs and quantifies the different contribution of each mechanism for different MNV patterns at the genome-wide scale.

**MNV distribution across different genomic regions.** We next examined how MNV pattern distributions differ between functional annotation categories. We used 13 different functional annotations such as coding sequence, enhancer, and promoter from Finucane et al.[41], and the DNA methylation annotation from the Encyclopedia of DNA Elements (ENCODE)[42], to calculate the number of MNVs that fall into each category (Supplementary Table 4). MNV density, defined as the number of MNVs observed in each functional category divided by the total length of the genomic interval belonging to each category, is shown in Fig. 4b and c. We found that MNV density of the substitution patterns typically involving CpG transitions is positively correlated with the methylation level (linear regression Pearson correlation $r = 0.95$ for CG->TA and $r = 0.87$ for CA->TG, $p < 10^{-3}$). Conversely, MNV density for non-CpG transversion-related substitution patterns, and the substitution patterns related to pol-zeta slippage, negatively correlates with methylation status (linear regression Pearson correlation $r = -0.90$ for GA->TC, $r = -0.91$ for AG->CC, $r = -0.91$ for GA->TT, and $r = -0.92$ for GC->AA, $p < 10^{-5}$; Fig. 4b, c).

Finally, we explored the effect of genic context on MNV origins and discovery: we selected the seven major regional annotations around gene-coding sequences[43,44], and calculated the fraction of MNVs likely explained by different mutational origins in each of these regions (Fig. 4d). Across all regions, we found that the MNV signal is primarily dominated by CpG transitions. The fraction of non-CpG transversions and polymerase slippage at repeats were consistently lower than (or nearly equal to) 5% of the overall signal. Pol-zeta signature was not as dominant as CpG transitions, except for at the transcription start site region, which has by far the lowest methylation rate in those seven annotations, and is thus expected to have a lower rate of CpG deamination mutations (which are dependent on the methylation of the original cytosine).

Overall, our results suggest that MNV density is highly dependent on the CpG methylation status of the surrounding sequence, and that MNVs that originate from non-CpG transversions or polymerase slippage at repeat junctions are relatively uncommon compared with those driven by CpG transitions or pol-zeta errors. Finally, MNVs that originate from pol-zeta error are the most common class of MNVs in the region close to the transcription start sites of genes, as low methylation levels in these regions result in low levels of CpG transitions.

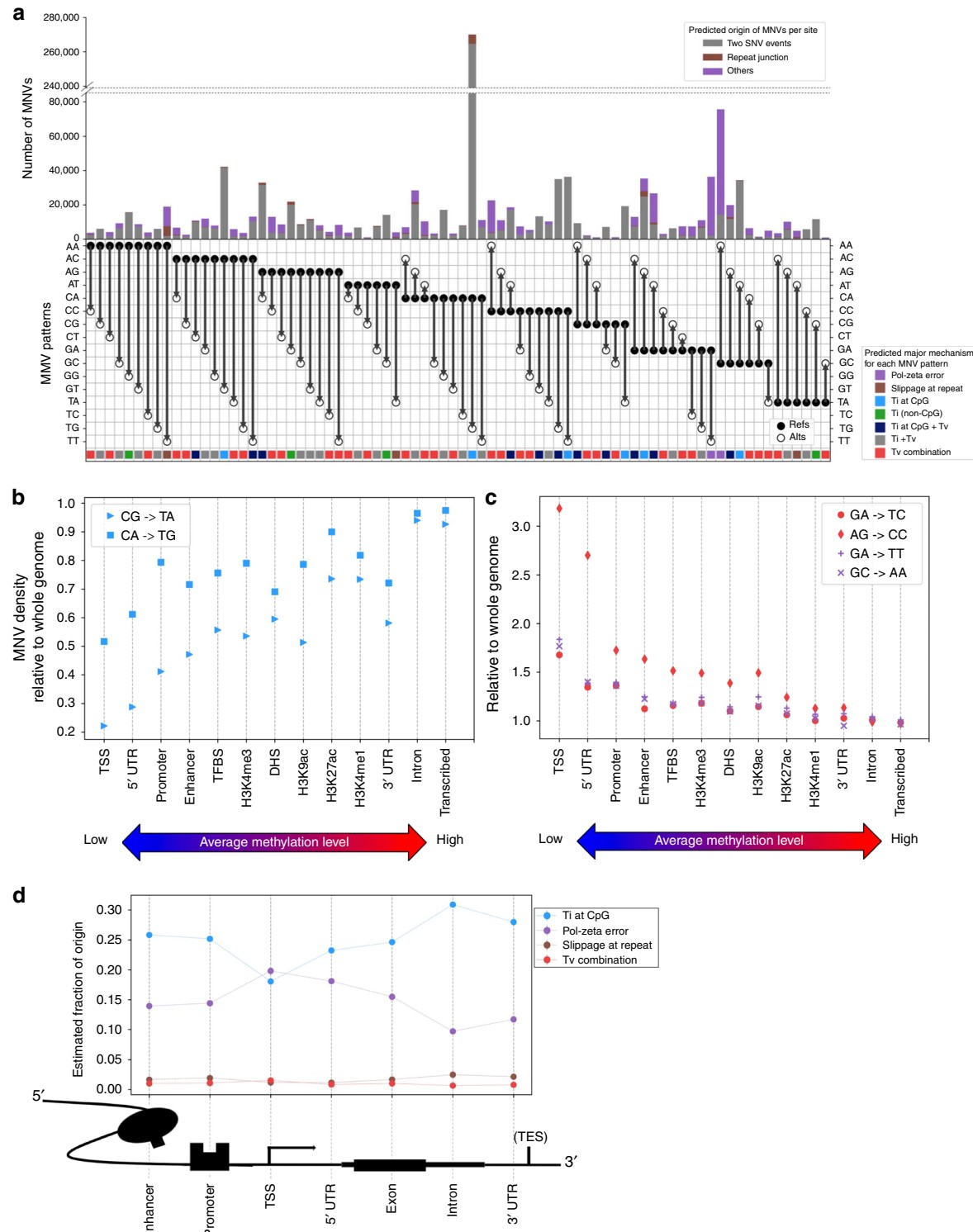

**Fig. 4** Distribution of MNVs across genome. **a** The number and the fraction of MNVs per origin, per substitution pattern. Gray are the estimated fraction of MNV originating from two single-nucleotide substitution events, brown for polymerase slippage at repeat contexts and purple are the others (presumably mainly replication error by pol-zeta). The colors along the bottom represent the estimated biological origins that dominate MNVs of that specific substitution pattern. **b**, **c** MNV density, defined as the number of MNVs per functional annotation divided by the base pair length in the annotation (relative to the whole-genome region), ordered by the methylation level of the functional category. **d** Estimated fraction of MNVs by different origins, per functional category around the coding region

## Discussion

We analyzed 125,748 human exomes and 15,708 genomes and identified 1,792,248 MNVs across genome with constituent variants falling within 2 bp distance, including 31,575 that exist within a codon. We have shown that MNVs represent an important class of genetic variation, and that they have a significant impact on the functional interpretation of genomic data, both at the population and individual level. Although we did not

encounter an individual in which an MNV is the likely cause of a rare disease after sequencing 6072 individuals from rare disease families, we expect that applying our pipeline to larger numbers of disease samples will identify previously missed diagnoses, as has been observed in another study of developmental delay cases[2].

The large number and high quality of variant calls in the gnomAD database provided increased power for statistical analysis of the three major mutational mechanisms (combinations of independent SNVs; replication errors by pol-zeta; and polymerase slippage at repeat junctions) responsible for the generation of MNVs, and importantly allowed us to estimate the relative contribution of each of these processes.

Our estimates of substitution pattern-specific MNV mutation rate and fraction come with important caveats. Our approach assumes that the local SNV mutation rate is invariant across instances of a specific 3 bp context; however, prior work has shown considerable regional variation in mutation rate across the genome, as well as variation driven by ancestry, environment, and other factors[45–48]. Another important limitation is the lack of confident estimates of insertion and deletion rate as a function of repeat length, which limits the confidence of our estimate of the fraction of polymerase slippage. Future large genome-scale data sets with more accurate insertion and deletion calls, likely involving long-read sequencing data, will be required to improve modeling of insertion and deletion mutations.

One clear feature of our data set was the signature of non-independence of mutational events separated by up to 10 bp, as suggested in various de novo studies[16,29,38–40]; further investigation of these clustered mutations, and contextualizing them with known sources of genomic instability, such as homologous recombination[49] or transposable elements[50,51], will be informative in exploring the mechanisms of clustered mutations.

The complete list of MNVs identified in gnomAD is publicly available (https://gnomad.broadinstitute.org/downloads), with the allele count annotated for both genome and exome. For the coding regions, we have also annotated the functional consequence of constituent SNVs and MNVs separately, and made the result viewable in an intuitive browser (https://gnomad.broadinstitute.org). Although some fraction of MNVs is missing from this list due to incomplete phasing sensitivity and read coverage, the database provides the most comprehensive set of estimates of MNV allele frequencies to date, valuable for further analysis of mutational mechanisms as well as the interpretation of MNVs in rare disease and cancer genomics[52,53].

Finally, despite the large sample size of our MNV data set, the fraction of MNVs that we have observed out of all the possible MNV configurations is still very far from saturating the space of possible MNVs, with only ~0.005% of all possible adjacent MNVs observed in our data (Supplementary Figs. 10, 11). Increasing the number of sequenced individuals[54] in both disease and non-disease cohorts will permit the discovery and determination of the phenotypic impact of an increasingly comprehensive catalogue of variation. This study confirms the importance of incorporating haplotypic phase into these efforts to permit the discovery and accurate interpretation of the full range of human variation.

## Methods

**Ethics**. We have complied with all relevant ethical regulations. This study was overseen by the Broad Institute's Office of Research Subject Protection and the Partners Human Research Committee, and was given a determination of Not Human Subjects Research. Informed consent was obtained from all participants.

**MNV calling**. 125,748 human exomes and 15,708 genomes from gnomAD 2.1 callset were used for the analyses (Supplementary Tables 5,6). We used hail (https://github.com/hail-is/hail), an open source, cloud-based scalable analysis tool for large genomic data. For MNV discovery, we exhaustively looked for variants

that appear in the same individual, in cis, and within 2 bp distance for the exome data set and 10 bp distance for the genome data set, using the hail *window_by_locus* function (i.e., we computationally checked every pair of genotypes within a certain window size, for every individual, to see whether the individual carries a pair(s) of mutation in the same haplotype. See Supplementary Methods for further detail. Also, we did not expand the window size >10 bp for MNV discovery, as phasing sensitivity significantly drops when the distance between variants is >10 bp, as shown in Supplementary Fig. 1d). For trio-based analyses, we expanded the range to 100 bp to obtain a more macroscopic view. Although we performed MNV calling in sex chromosomes for the coding region, we restricted our analysis to autosomes, in order to control for differences in zygosity.

MNV calling in rare disease samples was performed in a similar fashion as in the gnomAD exome data set. In total, 6072 rare disease whole-exome sequences were curated at the Broad Center for Mendelian Genomics (CMG)[55] and went through the MNV calling pipeline with the window size of 2 bp distance. The phenotypes observed in the cohort include: muscle disease such as Limb Girdle Muscular Dystrophy (LGMD; roughly one-third of the total), neurodevelopmental disorders, or severe phenotypes in eye, kidney, cardiac, or other orphan diseases (Supplementary Data 2).

**MNV filtering**. In the gnomAD MNV analysis, variant pairs for which one or both of their components have low quality reads were filtered out. Specifically, we only selected the variant sites that pass the Random Forest filtering, resulting in acceptance of 53.3% of the initial MNV candidates (Supplementary Fig. 12a). We also filtered out variant sites that are classified as low-complexity regions (LCRs) identified with the symmetric DUST algorithm[56] at a score threshold of 30, and additionally applied adjusted threshold criteria (GQ ≥ 20, DP ≥ 10, and allele balance > 0.2 for heterozygote genotypes) for filtering individual variants (Supplementary Table 7). For each MNV site, we annotated the number of alleles that appear as MNV, as well as the number of individuals carrying the MNV as a homozygous variant. The distribution of MNV sites that contain homozygous MNVs is shown in Supplementary Fig. 13. We also collapsed the MNV patterns that are reverse complements of each other, after observing that the number of MNVs are roughly symmetric (before collapsing, the ratio of each MNV pattern to its corresponding reverse complement pattern was mostly close to 1, with 0.95 being the lowest and 1.10 being the highest for adjacent MNVs) (Supplementary Fig. 14). All the MNV patterns in the main text and figures are equivalent to their reverse complement, and we do not distinguish them.

For the rare disease cohort, since our motivation was to find a definite example where an MNV is acting as a causal variant for a rare disease with severe phenotype rather than obtaining the population-level statistics, we did not apply site and sample-specific filtering, as opposed to the gnomAD MNV analysis. Instead of being computationally filtered by read quality, the 129 putative MNVs (16 gained nonsense mutations, 110 changed missense with high CADD score and low gnomAD MNV frequency, and 3 gained missense) went through manual inspection by the analysts at the Center for Mendelian Genomics (CMG) at the Broad Institute[55], after annotating the affected gene. Specifically, all the variants were checked manually under the criteria below:

- Whether the gene affected is constrained in the gnomAD population.
- Whether the case has already been solved with other causal variant.
- Whether the MNV looks real in the Interactive Genome Browser (IGV).[57]
- Whether the MNV is in the proband and, if applicable, the segregation pattern of the MNV
- Whether the known function of the gene affected matches the patient phenotype.

MNVs were filtered out if they failed one or more of the criteria above. These results suggest that MNVs explain only a small fraction of undiagnosed genetic disease cases, consistent with their overall frequency as a class of variation, and with prior work in large disease-affected cohorts[2]. The summary for MNV analysis in rare disease cohort is also available at Supplementary Data 2.

**Analysis of phasing sensitivity**. In order to compare the phasing information derived from different methods (read-based and trio-based), we took an approach of comparing the relative phase (binary classification of whether two SNVs of MNV are in the same haplotype or not), as shown in Supplementary Table 8. We investigated the heterozygous variant pairs whose phasing information is not provided by the trio-based phasing and observed that majority (83.5%) of the cases reflected both parents carrying a heterozygous variant, a scenario where trio-based phasing is inherently uninformative. We also investigated the heterozygous variant pairs whose phasing information is not provided by the read-based phasing. Specifically, unphased pairs tend to have either low- or high-read depth (odds ratio = 3.20, Fisher's exact test, $p < 10^{-100}$ for low, and odds ratio = 2.33, Fisher's exact test, $p < 10^{-100}$ for high-read depth; Supplementary Table 3), consistent with our previous understanding that an excess of reads can lead to involvement of erroneous reads and thus reduce the confidence of phasing of HaplotypeCaller[58] (as well as the lack of the number of reads reduces the calling rate). All the statistical tests are two-sided, throughout the paper.

**Analysis of functional impact in coding region.** We focused on the coding region of the canonical transcript of genes and examined the codon change and their consequence for all the MNVs that fall in a single codon (see Supplementary Tables 9,10 for the number of MNVs that spans across two codons). When comparing with population-level constraint, for each MNV, we annotated the constraint metric (LOEUF[22]) of the gene whose protein product is affected. For rescued nonsense mutations, we took only the ones are rescued in all the individuals with the component variants (i.e., we excluded the ones whose allele count of MNVs are not equal to the allele count of the SNV that introduces a nonsense mutation), resulting in 1538 out of 1821 rescued nonsense mutations. We next used Loss-Of-Function Transcript Effect Estimator (LOFTEE[22]) in order to exclude the nonsense mutations that are not likely to affect the protein function. This resulted in 371 high-confidence (HC) gained nonsense mutations and 1400 HC rescued nonsense mutations, which were used for the population-level constraint analysis. In addition, we stratified the gene sets by core essential/nonessential genes from CRISPR/Cas knockout experiments[59,60] as an orthogonal indicator of gene constraint (Supplementary Fig. 2).

We did not include and correct for MNVs consisting of three SNVs in a single codon in the analysis of functional impact in coding region, since the number and frequency of such MNVs are significantly low (228 in total, with 5 newly gained nonsense, but no re-rescued or re-gained nonsense; 0.220 in total per person). The full list of such MNVs are available as a separate file at: https://gnomad.broadinstitute.org/downloads.

Frame-restoring indel analysis was performed in a similar fashion. We used the gnomAD exome data set to call and filter the insertion/deletion pairs using the same filtering criteria (except for the fact that we did not restrict our analysis to cases where the frameshift effect would be rescued in all individuals), and focused on the canonical transcripts for the functional impact evaluation.

**Defining one-step MNVs and MNVs in repetitive contexts.** A one-step MNV was defined as a MNV for which the allele count of both SNVs that make up the MNV is the same and close to the allele count of the MNV itself. We also compared the allele count of constituent SNVs (AC1 and AC2) with the allele count of the corresponding MNV (AC_mnv), and observed that the majority of one-step MNVs we discovered have AC1_mnv divided by AC1 >0.9 (Supplementary Fig. 15). Therefore, we expect the false discovery rate of one-step MNVs (misclassifying the MNV whose AC1 and AC2 are equal just by chance) to be limited. The full distribution of all the allele counts, including per-population characterizations, are shown in Supplementary Fig. 16 and Supplementary Table 11.

Repetitive sequences are defined by taking the ±4 bp context of the MNV and setting the threshold manually, by looking at the distribution of repeat contexts around all the MNVs (Supplementary Figs. 17, 18). Specifically, a sequence is defined as repetitive if the number of dinucleotide repeat units > 1, for both reference and alternative ±4 bp context, and the number of dinucleotide repeat units > 2, for either reference or alternative ±4 bp context, and, for adjacent MNVs only, if the reference and/or alternative 2 bp are mononucleotide repeat, increase the threshold by one mononucleotide repeat unit.

Here, dinucleotide repeat unit is defined as the reference or the alternative allele itself (with the gap when d > 1 and counting the overlap). For example, the reference and alternative dinucleotide repeat counts for TATATAT -> TAAAAAT are both 3). The third criteria was added specifically for adjacent MNVs to adjust for counting the overlap more than once. This threshold was set so that the number of MNVs with equal or higher repeats would be <5% of the total, corresponding to two standard deviations away from the mean, and also because the estimated mutation rate in these repetitive contexts is likely to be orders of magnitude higher than the background MNV mutation rate originating from the combination of two SNV events[35–37].

**Calculating the proportion of MNVs per biological origin.** We calculated the proportion of MNV per biological origin by comparing the observed number of MNVs (that are not in repetitive contexts) with the expected number of MNV under single-nucleotide mutational model.

Specifically, if we simply hypothesize most of the MNV are combination of two single-nucleotide substitution events, we can estimate the relative probability of MNV event per substitution pattern. For example, probability of observing a CA to TG MNV in a single individual, single site ($p(CA \rightarrow TG)$) is proportional to $p(CA \rightarrow TA) \, p(TA \rightarrow TG) + p(CA \rightarrow CG) \, p(CG \rightarrow TG)$, and probability of TA to GC MNV ($p(TA \rightarrow GC)$) is proportional to $p(TA \rightarrow GA) \cdot p(GA \rightarrow GC) + p(TA \rightarrow TC) \cdot p(TC \rightarrow GC)$. Former equation involves the product of transition at CpG, while both term of the latter are product of transversion at non-CpG, which works as a reasonable explanation of the frequency difference of those two MNV patterns.

Using the same principle (and accounting for reference base pair frequency, population number and global SNV mutation rate defined by 3 bp context[26], we first constructed a null model of MNV distribution. In reality, this null model does not represent the real distribution we observe, due to biological mechanisms that introduce MNV. Therefore, we allowed additional factor $q$, that denotes the mutational event where two SNVs are introduced at the same time. For the example of $p(CA \rightarrow TG)$, we model this probability to be proportional to $l \, p(TA \rightarrow GA) \cdot p(GA \rightarrow GC) + p(TA \rightarrow TC) \cdot p(TC \rightarrow GC) + q(CA \rightarrow TG)$,

and try to estimate the $q$ term, which corresponds to the proportion of MNVs that are explained by non-SNV (and non-repeat) factor. Further details are explained in the Supplementary Methods (section "Models and assumptions for calculating the proportion of MNV per biological mechanism").

In addition, for each of MNV pattern, we annotated the predicted major mechanism for each MNV pattern in the following order:
1. Pol-zeta, for the patterns known as polymerase signature (GA- > TT and GC- > AA)
2. Repeat, for the patterns whose fraction of MNVs in repeat contexts are >10% (corresponding to two standard deviations away from the mean; AA- > TT, AT- > TA, and TA- > AT)
3. One of Ti at CpG, Ti, Ti at CpG + Tv, Ti + Tv, Tv combination, based on possible combinations of single-nucleotide mutational processes. For example, Ti at CpG is when transition in CpG combined with another transition can occur in the mutational processes (Supplementary Data 3).

**Estimation of the global MNV rate per substitution pattern.** In order to estimate the global MNV mutation rate for adjacent MNVs, as well as the mutation rate per MNV pattern, we first focused the number of one-step MNVs, assuming that there are no recurrent mutations and therefore the allele frequency of constituent SNVs are equal if and only if it originates from an MNV event in a single generation. In this section, we will simply write one-step MNV of distance 1 bp (i.e., adjacent) as MNV.

We then calculated the global MNV mutation rate under the Watterson estimator model, as in Kaplanis et al.[2]. Specifically, we divided the number of MNV sites by the number of SNV sites in our gnomAD data set, and scaled by the global single-nucleotide mutation rate identified in previous research ($1.2 \times 10^{-8}$), which yielded $2.94 \times 10^{-11}$ per 2 bp per generation. This is roughly two-thirds of the estimation provided by the Kaplanis et al.[2] using trio data, slightly smaller presumably due to differing filtering method. Next, In order to get the mutation rate per 2 bp for each of the MNV patterns, we simply scaled the global MNV mutation rate described above by the number of reference 2 bp and the coverage difference. The full data for all the 78 patterns are shown in Supplementary Data 3. Further details are explained in the Supplementary Methods (section "Models and assumptions for estimation of the global MNV rate per substitution pattern").

**Functional enrichment.** Thirteen functional annotations were collected from Finucane et al.[41] as a bed file (which originates from database, such as ENCODE, Roadmap[61] and UCSC genome browser[62].) For the methylation data, we collected the genome methylation level from ENCODE, and calculated the fraction of methylated CpG out of all the CpGs in the region, and ordered by the fraction (Supplementary Table 4).

MNV density calculation was performed under the null hypothesis that the number of MNV of type WX→YZ we observe in an arbitrary genomic interval is proportional to the number of WX in the interval. Specifically, the MNV density of WX→YZ in interval $I$ is defined as

$$D(WX \rightarrow YZ|I) = \frac{N(WX \rightarrow YZ|I)}{N(WX|I)},$$ where $N(WX \rightarrow YZ|I)$ is the number of MNVs of WX→YZ, and $N(WX|I)$ is the number of WX in the reference genome we observe in that specific genomic interval. We then normalized the density by dividing by $D$ (WX→YZ|I = whole genome) for scaling purpose (i.e., $D(WX \rightarrow YZ|I) = k$ means that the probability of observing a mutation of WX→YZ given a sequence context of WX is $k$ times higher in genomic functional category $I$ than the overall genome.) For estimating the fraction of MNVs per origin, we took a thresholding approach and defined four MNVs (CA- > TG, AC- > GT, CC- > TT, and GA- > AG) as CpG signal, two (GC- > AA, GA- > TT) as pol-zeta, three as repeat (AA- > TT, TA- > AT, AT- > TA) and six transversion (TA- > GC, CG- > AT, AT- > CG, CG- > GC, GC- > CG, CG- > AC) signal (and left all the other 78-(4 + 2 + 3 + 6) = 63 patterns as others, in order to highlight the strongest signals) based on the result from Fig. 3. The fraction of MNVs per origin is then defined simply as the number of MNVs that fall into that pattern divided by all the MNVs, in the genomic interval. The coverage difference per interval was as small as negligible (Supplementary Table 4).

**Reporting summary.** Further information on research design is available in the Nature Research Reporting Summary linked to this article.

## Data availability

The list of coding MNVs in gnomAD exome are available at gs://gnomad-public/release/2.1/mnv/gnomad_mnv_coding.tsv (tab separated file). The coding MNVs consisting of three SNVs in a single codon is available as a separate file at gs://gnomad-public/release/2.1/mnv/gnomad_mnv_coding_3bp.tsv. The list of frame-restoring indel pairs are available at gs://gnomad-public/release/2.1/mnv/frame_restoring_indels.tsv. The list of all the MNVs in gnomAD genomes are available at gs://gnomad-public/release/2.1/mnv/genome/gnomad_mnv_genome_d{i}.tsv.bgz (tab separated file, compressed. Replace {i} (0 < i < 11) with the distance between two SNVs of MNV.), or gs://gnomad-public/release/2.1/mnv/genome/gnomad_mnv_genome_d{i}.ht (hail table. Replace {i} (0 < i < 11) with the distance between two SNVs of MNV.). Explanations for each column in each file can be found at gs://gnomad-public/release/2.1/mnv/mnv_readme.md. All the

files above are also available at the download page of the gnomAD browser (https://gnomad.broadinstitute.org/downloads).

## Code availability

The code used in the study is available at https://github.com/macarthur-lab/gnomad_mnv.

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

## Acknowledgements

We would like to thank the many individuals whose sequence data are aggregated in gnomAD for their contributions to research, and for making this work possible. The results published here are in part based upon data: (1) generated by The Cancer Genome Atlas managed by the NCI and NHGRI (accession: phs000178.v10.p8). Information about TCGA can be found at http://cancergenome.nih.gov, (2) generated by the Genotype-Tissue Expression Project (GTEx) managed by the NIH Common Fund and NHGRI (accession: phs000424.v7.p2), (3) generated by the Exome Sequencing Project, managed by NHLBI, (4) generated by the Alzheimer's Disease Sequencing Project (ADSP), managed by the NIA and NHGRI (accession: phs000572.v7.p4). We would like to thank the Hail team for developing tools essential for the large-scale computation in this work. We would like to thank the analysis team of the Broad's Rare Disease Group for their manual inspection of MNVs in rare disease cohorts. This work was funded by NIDDK U54 DK105566, NIGMS R01 GM104371, and NHGRI UM1 HG008900-01. Q. W. was supported by the Nakajima Foundation Scholarship. K.J.K. was supported by NIGMS F32 GM115208. A.O.D.L. was supported by NICHD K12 HD052896.

## Author contributions

Q.W. conducted the study, performed the analysis, and wrote the paper. E.P.H., A.J.H., and B.B.C. defined the MNV classification and drafted the research. A.O.D.L. provided the data set for rare disease analysis. L.C.F. and L.D.G. generated the trio-based and read-based phasing information. J.A., B.B.C., and K.J.K. reviewed and edited the paper. D.G. M. conceived the project, supervised the overall work, reviewed and edited the paper.

## Competing interests

D.G.M. is a founder with equity in Goldfinch Bio, and has received research support from AbbVie, Astellas, Biogen, BioMarin, Eisai, Merck, Pfizer, and Sanofi-Genzyme. K.J. K. owns stock in Personalis. E.V.M. has received research support in the form of charitable contributions from Charles River Laboratories and Ionis Pharmaceuticals, and has consulted for Deerfield Management. M.I.M.: The views expressed in this article are those of the author(s) and not necessarily those of the NHS, the NIHR, or the Department of Health. He has served on advisory panels for Pfizer, NovoNordisk, Zoe Global; has received honoraria from Merck, Pfizer, NovoNordisk, and Eli Lilly; has stock options in Zoe Global and has received research funding from Abbvie, Astra Zeneca, Boehringer Ingelheim, Eli Lilly, Janssen, Merck, NovoNordisk, Pfizer, Roche, Sanofi Aventis, Servier, and Takeda. As of June 2019, M.I.M. is an employee of Genentech, and holds stock in Roche. R.K.W. has received unrestricted research grants from Takeda Pharmaceutical Company. M.J.D. is a founder of Maze Therapeutics. B.M.N. is a member of the scientific advisory board at Deep Genomics and consultant for Camp4 Therapeutics, Takeda Pharmaceutical, and Biogen. A.O.D.L. has received honoraria from ARUP and Chan Zuckerberg Initiative.

## Additional information

## Genome Aggregation Database Production Team

Irina M. Armean[1,2,9], Eric Banks[5], Louis Bergelson[5], Kristian Cibulskis[5], Ryan L. Collins[1,3,10], Kristen M. Connolly[11], Miguel Covarrubias[5], Mark J. Daly[1,2,12], Stacey Donnelly[1], Yossi Farjoun[5], Steven Ferriera[13], Stacey Gabriel[13], Jeff Gentry[5], Namrata Gupta[1,13], Thibault Jeandet[5], Diane Kaplan[5], Kristen M. Laricchia[1,2], Christopher Llanwarne[5], Eric V. Minikel[1], Ruchi Munshi[5], Benjamin M. Neale[1,2], Sam Novod[5], Nikelle Petrillo[5], Timothy Poterba[1,2,12], David Roazen[5], Valentin Ruano-Rubio[5], Andrea Saltzman[1], Kaitlin E. Samocha[14], Molly Schleicher[1], Cotton Seed[2,12], Matthew Solomonson[1,2], Jose Soto[5], Grace Tiao[1,2], Kathleen Tibbetts[5], Charlotte Tolonen[5], Christopher Vittal[2,12], Gordon Wade[5], Arcturus Wang[1,2,12], James S. Ware[1,15,16], Nicholas A. Watts[1,2], Ben Weisburd[5] & Nicola Whiffin[1,15,16]

[9]European Molecular Biology Laboratory, European Bioinformatics Institute, Wellcome Genome Campus, Hinxton, Cambridge CB10 1SD, UK. [10]Center for Genomic Medicine, Massachusetts General Hospital, Boston, MA 02114, USA. [11]Genomics Platform, Broad Institute of MIT and Harvard, Cambridge, MA 02142, USA. [12]Stanley Center for Psychiatric Research, Broad Institute of MIT and Harvard, Cambridge, MA 02142, USA. [13]Broad Genomics, Broad Institute of MIT and Harvard, Cambridge, MA 02142, USA. [14]Wellcome Sanger Institute, Wellcome Genome Campus, Hinxton, Cambridge CB10 1SA, UK. [15]National Heart & Lung Institute and MRC London Institute of Medical Sciences, Imperial College London, London W12 0NN, UK. [16]Cardiovascular Research Centre, Royal Brompton & Harefield Hospitals NHS Trust, London SW3 6NP, UK

## Genome Aggregation Database Consortium

Carlos A. Aguilar Salinas[17], Tariq Ahmad[18], Christine M. Albert[19,20], Diego Ardissino[21], Gil Atzmon[22,23], John Barnard[24], Laurent Beaugerie[25], Emelia J. Benjamin[26,27,28], Michael Boehnke[29], Lori L. Bonnycastle[30], Erwin P. Bottinger[31], Donald W. Bowden[32,33,34], Matthew J. Bown[35,36], John C. Chambers[37,38,39], Juliana C. Chan[40], Daniel Chasman[19,41], Judy Cho[31], Mina K. Chung[42], Bruce Cohen[41,43], Adolfo Correa[44], Dana Dabelea[45], Dawood Darbar[46], Ravindranath Duggirala[47], Josée Dupuis[26,48], Patrick T. Ellinor[1,49], Roberto Elosua[50,51,52], Jeanette Erdmann[53,54,55], Tõnu Esko[1,56], Martti Färkkilä[57], Jose Florez[1,10,41], Andre Franke[58], Gad Getz[41,59,60], Benjamin Glaser[61], Stephen J. Glatt[62], David Goldstein[63,64], Clicerio Gonzalez[65], Leif Groop[66,67], Christopher Haiman[68], Craig Hanis[69], Matthew Harms[64,70], Mikko Hiltunen[71], Matti M. Holi[72], Christina M. Hultman[73,74], Mikko Kallela[75], Jaakko Kaprio[68,76], Sekar Kathiresan[1,10,41], Bong-Jo Kim[77], Young Jin Kim[77], George Kirov[78], Jaspal Kooner[38,39,79], Seppo Koskinen[80], Harlan M. Krumholz[81,82], Subra Kugathasan[83], Soo Heon Kwak[84], Markku Laakso[85,86], Terho Lehtimäki[87], Ruth J.F. Loos[31,88], Steven A. Lubitz[1,50], Ronald C.W. Ma[40,89,90], Jaume Marrugat[52,91], Kari M. Mattila[87], Steven McCarroll[10,92], Mark I. McCarthy[93,94,95], Dermot McGovern[96], Ruth McPherson[97], James B. Meigs[1,41,98], Olle Melander[99], Andres Metspalu[56], Peter M. Nilsson[100], Michael C. O'Donovan[78], Dost Ongur[41,43], Lorena Orozco[101], Michael J. Owen[78], Colin N.A. Palmer[102], Aarno Palotie[1,12,67], Kyong Soo Park[84,103], Carlos Pato[104], Ann E. Pulver[105], Nazneen Rahman[106], Anne M. Remes[107], John D. Rioux[108,109], Samuli Ripatti[1,67,76], Dan M. Roden[110,111], Danish Saleheen[112,113,114], Veikko Salomaa[115], Nilesh J. Samani[35,36], Jeremiah Scharf[1,10,12], Heribert Schunkert[116,117], Moore B. Shoemaker[118], Pamela Sklar[119,120,121], Hilkka Soininen[122], Harry Sokol[25], Tim Spector[123], Patrick F. Sullivan[75,124], Jaana Suvisaari[115], E. Shyong Tai[125,126,127], Yik Ying Teo[125,128,129], Tuomi Tiinamaija[67,130,131], Ming Tsuang[132,133], Dan Turner[133,134], Teresa Tusie-Luna[135,136], Erkki Vartiainen[76], Hugh Watkins[137], Rinse K. Weersma[138], Maija Wessman[67,130], James G. Wilson[139] & Ramnik J. Xavier[140,141]

[17]Unidad de Investigacion de Enfermedades Metabolicas. Instituto Nacional de Ciencias Medicas y Nutricion, Mexico City 14080, Mexico. [18]Peninsula College of Medicine and Dentistry, Exeter EX25DW, UK. [19]Division of Preventive Medicine, Brigham and Women's Hospital, Boston, MA 02115, USA. [20]Division of Cardiovascular Medicine, Brigham and Women's Hospital and Harvard Medical School, Boston, MA 02115, USA. [21]Department of Cardiology, University Hospital, 43100 Parma, Italy. [22]Department of Biology, Faculty of Natural Sciences, University of Haifa, Haifa 3498838, Israel. [23]Departments of Medicine and Genetics, Albert Einstein College of Medicine, Bronx, NY 10461, USA. [24]Department of Quantitative Health Sciences, Lerner Research Institute, Cleveland Clinic, Cleveland, OH 44122, USA. [25]Sorbonne Université, APHP, Gastroenterology Department, Saint Antoine Hospital, Paris 75012, France. [26]NHLBI and Boston University's Framingham Heart Study, Framingham, MA 01702, USA. [27]Department of Medicine, Boston University School of Medicine, Boston, MA 02118, USA. [28]Department of Epidemiology, Boston University School of Public Health, Boston, MA 02118, USA. [29]Department of Biostatistics and Center for Statistical Genetics, University of Michigan, Ann Arbor, MI 48109, USA. [30]National Human Genome Research Institute, National Institutes of Health, Bethesda, MD 20892, USA. [31]The Charles Bronfman Institute for Personalized Medicine, Icahn School of Medicine at Mount Sinai, New York, NY 10029, USA. [32]Department of Biochemistry, Wake Forest School of Medicine, Winston-Salem, NC 27101, USA. [33]Center for Genomics and Personalized Medicine Research, Wake Forest School of Medicine, Winston-Salem, NC 27157, USA. [34]Center for Diabetes Research, Wake Forest School of Medicine, Winston-Salem, NC 27101, USA. [35]Department of Cardiovascular Sciences, University of Leicester, Leicester LE1 7RH, UK. [36]NIHR Leicester Biomedical Research Centre, Glenfield Hospital, Leicester LE3 9QP, UK. [37]Department of Epidemiology and Biostatistics, Imperial College London, London W2 1PG, UK. [38]Department of Cardiology, Ealing Hospital NHS Trust, Southall UB1 3HW, UK. [39]Imperial College Healthcare NHS Trust, Imperial College London, London W2 1NY, UK. [40]Department of Medicine and Therapeutics, The Chinese University of Hong Kong, Hong Kong, China. [41]Department of Medicine, Harvard Medical School, Boston, MA 02115, USA. [42]Departments of Cardiovascular Medicine, Cellular and Molecular Medicine, Molecular Cardiology and Quantitative Health Sciences, Cleveland Clinic, Cleveland, OH 44195, USA. [43]McLean Hospital, Belmont, MA 02478, USA. [44]Department of Medicine, University of Mississippi Medical Center, Jackson, MS 39216, USA. [45]Department of Epidemiology, Colorado School of Public Health, Aurora, CO 80045, USA. [46]Department of Medicine and Pharmacology, University of Illinois at Chicago, Chicago, IL 60612, USA. [47]Department of Genetics, Texas Biomedical Research Institute, San Antonio, TX 78227, USA. [48]Department of Biostatistics, Boston University School of Public Health, Boston, MA 02118, USA. [49]Cardiac Arrhythmia Service and Cardiovascular Research Center, Massachusetts General Hospital, Boston, MA 02114, USA. [50]Cardiovascular Epidemiology and Genetics, Hospital del Mar Medical Research Institute (IMIM), Barcelona 08003 Catalonia, Spain. [51]CIBER CV, Barcelona 08017 Catalonia, Spain. [52]Department of Medicine, Medical School, University of Vic-Central University of Catalonia, Barcelona 08500, Spain. [53]Institute for Cardiogenetics, University of Lübeck, Lübeck 23562, Germany. [54]DZHK (German Research Centre for Cardiovascular Research), Partner Site Hamburg/Lübeck/Kiel, 23562 Lübeck, Germany. [55]University Heart Center Lübeck, 23562 Lübeck, Germany. [56]Estonian Genome Center, Institute of Genomics, University of Tartu, Tartu 51003, Estonia. [57]Helsinki University and Helsinki University Hospital, Clinic of Gastroenterology, Helsinki 00100, Finland. [58]Institute of Clinical Molecular

Biology (IKMB), Christian-Albrechts-University of Kiel, Kiel 24118, Germany. [59]Bioinformatics Program, MGH Cancer Center and Department of Pathology, Boston, MA 02129, USA. [60]Cancer Genome Computational Analysis, Broad Institute, Cambridge, MA 02142, USA. [61]Endocrinology and Metabolism Department, Hadassah-Hebrew University Medical Center, Jerusalem 91120, Israel. [62]Department of Psychiatry and Behavioral Sciences, SUNY Upstate Medical University, Oneida, NY 13421, USA. [63]Institute for Genomic Medicine, Columbia University Medical Center, Hammer Health Sciences, 1408, 701 West 168th Street, New York, NY 10032, USA. [64]Department of Genetics & Development, Columbia University Medical Center, Hammer Health Sciences, 1602, 701 West 168th Street, New York, NY 10032, USA. [65]Centro de Investigacion en Salud Poblacional. Instituto Nacional de Salud Publica MEXICO, Mexico 62100, Mexico. [66]Lund University, Lund SE-221 00, Sweden. [67]Institute for Molecular Medicine Finland (FIMM), HiLIFE, University of Helsinki, Helsinki 00014, Finland. [68]Lund University Diabetes Centre, Lund SE-214 28, Sweden. [69]Human Genetics Center, University of Texas Health Science Center at Houston, Houston, TX 77030, USA. [70]Department of Neurology, Columbia University, New York, NY 10032, USA. [71]Institute of Biomedicine, University of Eastern Finland, Kuopio 70210, Finland. [72]Department of Psychiatry, PL 320, Helsinki University Central Hospital, Lapinlahdentie, 00 180 Helsinki, Finland. [73]Department of Medical Epidemiology and Biostatistics, Karolinska Institutet, Stockholm 171 77, Sweden. [74]Icahn School of Medicine at Mount Sinai, New York, NY 10029, USA. [75]Department of Neurology, Helsinki University Central Hospital, Helsinki 00290, Finland. [76]Department of Public Health, Faculty of Medicine, University of Helsinki, Helsinki 00014, Finland. [77]Center for Genome Science, Korea National Institute of Health, Chungcheongbuk-do 363-951, Republic of Korea. [78]MRC Centre for Neuropsychiatric Genetics & Genomics, Cardiff University School of Medicine, Hadyn Ellis Building, Maindy Road, Cardiff CF24 4HQ, UK. [79]National Heart and Lung Institute, Cardiovascular Sciences, Hammersmith Campus, Imperial College London, London SW3 6LY, UK. [80]Department of Health, THL-National Institute for Health and Welfare, 00271 Helsinki, Finland. [81]Section of Cardiovascular Medicine, Department of Internal Medicine, Yale School of Medicine, New Haven, CT 06510, USA. [82]Center for Outcomes Research and Evaluation, Yale-New Haven Hospital, New Haven, CT 06510, USA. [83]Division of Pediatric Gastroenterology, Emory University School of Medicine, Atlanta, GA 30322, USA. [84]Department of Internal Medicine, Seoul National University Hospital, Seoul 03080, Republic of Korea. [85]Institute of Clinical Medicine, The University of Eastern Finland, Kuopio 70210, Finland. [86]Kuopio University Hospital, Kuopio 70210, Finland. [87]Department of Clinical Chemistry, Fimlab Laboratories and Finnish Cardiovascular Research Center-Tampere, Faculty of Medicine and Health Technology, Tampere University, Tampere 33720, Finland. [88]The Mindich Child Health and Development Institute, Icahn School of Medicine at Mount Sinai, New York, NY 10029, USA. [89]Li Ka Shing Institute of Health Sciences, The Chinese University of Hong Kong, Hong Kong, China. [90]Hong Kong Institute of Diabetes and Obesity, The Chinese University of Hong Kong, Hong Kong, China. [91]Cardiovascular Research REGICOR Group, Hospital del Mar Medical Research Institute (IMIM), Barcelona 08003 Catalonia, Spain. [92]Department of Genetics, Harvard Medical School, Boston, MA 02115, USA. [93]Oxford Centre for Diabetes, Endocrinology and Metabolism, University of Oxford, Churchill Hospital, Old Road, Headington, Oxford OX3 7LJ, UK. [94]Wellcome Centre for Human Genetics, University of Oxford, Roosevelt Drive, Oxford OX3 7BN, UK. [95]Oxford NIHR Biomedical Research Centre, Oxford University Hospitals NHS Foundation Trust, John Radcliffe Hospital, Oxford OX3 9DU, UK. [96]F Widjaja Foundation Inflammatory Bowel and Immunobiology Research Institute, Cedars-Sinai Medical Center, Los Angeles, CA 90048, USA. [97]Atherogenomics Laboratory, University of Ottawa Heart Institute, Ottawa ON K1Y 4W7, Canada. [98]Division of General Internal Medicine, Massachusetts General Hospital, Boston, MA 02114, USA. [99]Department of Clinical Sciences, University Hospital Malmo Clinical Research Center, Lund University, Malmo 205 02, Sweden. [100]Lund University, Department of Clinical Sciences, Skane University Hospital, Malmo 222 42, Sweden. [101]Instituto Nacional de Medicina Genómica (INMEGEN), Mexico City 14610, Mexico. [102]Medical Research Institute, Ninewells Hospital and Medical School, University of Dundee, Dundee DD1 9SY, UK. [103]Department of Molecular Medicine and Biopharmaceutical Sciences, Graduate School of Convergence Science and Technology, Seoul National University, Seoul 08826, Republic of Korea. [104]Department of Psychiatry, Keck School of Medicine at the University of Southern California, Los Angeles, CA 90033, USA. [105]Department of Psychiatry and Behavioral Sciences, Johns Hopkins University School of Medicine, Baltimore, MD 21205, USA. [106]Division of Genetics and Epidemiology, Institute of Cancer Research, London SM2 5NG, UK. [107]Medical Research Center, Oulu University Hospital, Oulu, Finland and Research Unit of Clinical Neuroscience, Neurology, University of Oulu, Oulu 90014, Finland. [108]Research Center, Montreal Heart Institute, Montreal, Quebec H1T 1C8, Canada. [109]Department of Medicine, Faculty of Medicine, Université de Montréal, Québec H3T 1J4, Canada. [110]Department of Biomedical Informatics, Vanderbilt University Medical Center, Nashville, TN 37212, USA. [111]Department of Medicine, Vanderbilt University Medical Center, Nashville, TN 37212, USA. [112]Department of Biostatistics and Epidemiology, Perelman School of Medicine at the University of Pennsylvania, Philadelphia, PA 19104, USA. [113]Department of Medicine, Perelman School of Medicine at the University of Pennsylvania, Philadelphia, PA 19104, USA. [114]Center for Non-Communicable Diseases, Karachi 75300, Pakistan. [115]National Institute for Health and Welfare, Helsinki 00271, Finland. [116]Deutsches Herzzentrum München, München 80636, Germany. [117]Technische Universität München, München 80333, Germany. [118]Division of Cardiovascular Medicine, Nashville VA Medical Center and Vanderbilt University, School of Medicine, Nashville, TN 37232-8802, USA. [119]Department of Psychiatry, Icahn School of Medicine at Mount Sinai, New York, NY 10029, USA. [120]Department of Genetics and Genomic Sciences, Icahn School of Medicine at Mount Sinai, New York, NY 10029, USA. [121]Institute for Genomics and Multiscale Biology, Icahn School of Medicine at Mount Sinai, New York, NY 10029, USA. [122]Institute of Clinical Medicine, Neurology, University of Eastern Finland, Kuopio 80101, Finland. [123]Department of Twin Research and Genetic Epidemiology, King's College London, London WC2R 2LS, UK. [124]Departments of Genetics and Psychiatry, University of North Carolina, Chapel Hill, NC 27599, USA. [125]Saw Swee Hock School of Public Health, National University of Singapore, National University Health System, Singapore 117549, Singapore. [126]Department of Medicine, Yong Loo Lin School of Medicine, National University of Singapore, Singapore, Singapore. [127]Duke-NUS Graduate Medical School, Singapore 169857, Singapore. [128]Life Sciences Institute, National University of Singapore, Singapore 117456, Singapore. [129]Department of Statistics and Applied Probability, National University of Singapore, Singapore 117546, Singapore. [130]Folkhälsan Institute of Genetics, Folkhälsan Research Center, Helsinki 00250, Finland. [131]HUCH Abdominal Center, Helsinki University Hospital, Helsinki 00100, Finland. [132]Center for Behavioral Genomics, Department of Psychiatry, University of California, San Diego, CA 92093, USA. [133]Institute of Genomic Medicine, University of California, San Diego, CA 92093, USA. [134]Juliet Keidan Institute of Pediatric Gastroenterology, Shaare Zedek Medical Center, The Hebrew University of Jerusalem, Jerusalem 91905, Israel. [135]Instituto de Investigaciones Biomédicas UNAM, Mexico City 04510, Mexico. [136]Instituto Nacional de Ciencias Médicas y Nutrición Salvador Zubirán Mexico City, Mexico City 14080, Mexico. [137]Radcliffe Department of Medicine, University of Oxford, Oxford OX3 9DU, UK. [138]Department of Gastroenterology and Hepatology, University of Groningen and University Medical Center Groningen, Groningen 9713, The Netherlands. [139]Department of Physiology and Biophysics, University of Mississippi Medical Center, Jackson, MS 39216, USA. [140]Program in Infectious Disease and Microbiome, Broad Institute of MIT and Harvard, Cambridge, MA 02142, USA. [141]Center for Computational and Integrative Biology, Massachusetts General Hospital, Boston, MA 02114, USA

