## [Peer Review File · Nature Communications]

Reviewers' Comments:

Reviewer #1:

Remarks to the Author:

This manuscript from the gnomAD group addresses the topic of multi-nucleotide variants (MNVs) present in their large collection of whole genomes and exomes. As they have shown in other publications, their database is one of the largest freely available databases of human genome variation. Most variant callers are geared towards calling single nucleotide variants (SNPs/SNVs), and some variant callers explicitly remove MNVs because they are a signal of low quality reads due to there being multiple mismatches from the reference within a short distance. For this reason, the calling of MNVs is more challenging and requires the adaptation of algorithms. Overall, this is a very thorough manuscript, that includes a very large amount of data, but there are certain points that would improve it and make it better.

1. A large amount of the text is listing statistics concerning the MNVs that were found. These paragraphs are difficult to read and the data might be better presented either as tables or as plots.
2. In general usage, there is a distinction between SNP and SNV concerning the prevalence of a variant in the population. The authors should comment on this naming for multinucleotide variants and state which should be viewed as MNPs and which as MNVs
3. The limit of 10bp for the phasing analysis and the MNV analysis seems a bit arbitrary, especially when the reads are at least 100bp in size. This cutoff should be justified, or the authors should go up to collecting phased variants over longer haplotypes and analyze them.
4. How often are there more than 2 variants within 10bp? This should be explicitly noted. If the phasing results are sufficient, it would be nice to see this over longer distances.
5. It is noted that only ~0.005% of all possible MNVs were detected in this study. Presumably if the entire population of the world were sequenced, we still would not get every possible MNV across the genome.
6. Instead of just considering the functional impact of MNVs within one codon, it would be interesting to know how much of an impact all MNVs produce regardless of whether they are within one codon or across multiple codons. This is a more difficult computational task, but the real question of altered function is of a protein as a whole, not just an individual codon.

Overall, this manuscript represents a great contribution to the field and these minor changes will improve it.

Jeffrey A. Rosenfeld, Ph. D
Rutgers Cancer Institute of New Jersey

Reviewer #2:

Remarks to the Author:

In this manuscript, the authors examined a large dataset (125,748 human exomes and 15,708 genomes) for multi-nucleotide variants, a class of variants seldom used in disease gene mutation analysis. They made the obvious case that the effect of two variants found in the same codon would have very different effects on the amino acid the reference codon encoded than when each variant was considered alone. With a large dataset, they provide additional evidence for the mechanisms that give rise to MNVs. While none of the analyses done are novel, the comprehensive catalog is a good contribution to our understanding of genome diversity. There are several points that the authors should address:

1. The authors made the following statement, "the most frequent MNV pattern is CA->TG substitutions, which are likely to occur as a combination of an A->G transition, followed by a high mutation rate C->T CpG transition." If this two-step process indeed occurs in this particular order,

they should be able to find samples with higher frequency of CG (A->G occurring first) but not TA (assuming not everyone has the second mutation given a large sample size) at the CA->TG sites. Is this indeed the case?

2. "To investigate the extent of pol-zeta signature, we calculated the number of MNVs in which the gnomAD allele counts of the constitutive single nucleotide variants are equal (following previous methodology²), and observed that these "one-step" MNVs are significantly enriched in MNV patterns matching the pol-zeta signature..." Why not show that all samples either have WT-WT or ALT-ALT but not WT-ALT (WT-ALT would indicate a two-step process)?

3. Finding MNVs in repetitive sequence content is going to be problematic because misalignment can definitely contribute to this. How many of the total MNVs reside within repetitive regions (about 10%?).

4. The axes for figure 3b-d are slightly confusing when they switch from number to fraction in panel c and d. Best to stick with numbers for consistency.

5. How many of the 1,996,125 MNVs with constituent variants fall within 2 bp of each other are singletons (seen only in one sample)? What is the "allele frequency" distribution of these MNVs? Similarly, of the 6,261,326 MNVs with constituent variants fall within 10 bp of each other, how many are singletons and what is the "allele frequency" distribution? How many MNVs on average are found in one exome? In one genome? Are there ethnic differences? This information will help the readers understand the scope of this variant class.

6. I am surprised that the phasing sensitivity is only 85% for adjacent heterozygous variant pairs. Since exomes and genomes are typically with >30X coverage, there must be enough high quality reads that cover adjacent variants. An explanation of the low sensitivity is needed.

**Landscape of multi-nucleotide variants in 125,748 human exomes and 15,708 genomes
(Wang et al, NCOMMS-19-10112-T)
Response to reviewers**

We thank the reviewers for their thoughtful comments and have made a number of changes to the manuscript in response, which we believe have substantially improved this work. In particular we wish to note that a new filter for highly repetitive regions was applied across all of our variants in response to reviewer 2 (point 3 below), which slightly alters all of the numbers and figures in the text, but we believe makes our overall findings more robust to potential mapping artifacts.

Response to Reviewer 1:

1. A large amount of the text is listing statistics concerning the MNVs that were found. These paragraphs are difficult to read and the data might be better presented either as tables or as plots.

Readers can find basic information in Supplementary Table 3, and Figures 2 and 3 describe most of the numbers that are important in the manuscript. However, we agree that there are some numbers that are difficult to extract from these display items. In order to make these clearer, we have added another subsection that summarizes the numbers as Supplementary Table 3b.

Table S3. Summary of the study

b, Summary of the numbers of different MNVs in this paper

Exome	within codon	pairs of 2 SNVs: 31,575	Changes codon interpretation: 18,756
			Does not change: 12,819
		combination of 3 SNVs: 228	
	spanning 2 codons	pairs of 2 SNVs: 23,429	
Genome	adjacent MNVs	1,223,006	one-step MNV: 488,236
			In repetitive context: 38,504
			Most common (CA->TG): 270,071
			Least common (TA->GC): 988
	distance < 3	1,792,248	
	distance < 11	5,513,219	

2. In general usage, there is a distinction between SNP and SNV concerning the prevalence of a variant in the population. The authors should comment on this naming for multinucleotide variants and state which should be viewed as MNPs and which as MNVs

Although we understand that some researchers make a distinction between SNP and SNV, depending on the frequency of the variant, it has been difficult to find consensus on the frequency threshold used for this nomenclature. Therefore, for the sake of consistency, we use the more general term multi-nucleotide “variant” instead of “polymorphism” throughout the paper.

3. The limit of 10bp for the phasing analysis and the MNV analysis seems a bit arbitrary, especially when the reads are at least 100bp in size. This cutoff should be justified, or the authors should go up to collecting phased variants over longer haplotypes and analyze them.

There is a significant drop in the fraction of phased heterozygous pairs of variants when the distance is larger than 10bp, as shown in Supplementary Figure 1d. Phasing over longer haplotypic regions is definitely an interesting area to explore, but falls out of this paper’s scope, as it requires methods other than read-based phasing. To clarify the rationale for this cut-off we added a sentence in the method section (“Also, we did not expand the window size larger than 10 bp for the MNV discovery, as the phasing sensitivity significantly drops when the distance between variants is larger than 10 bp, as shown in Supplementary Fig. 1d”) and supplementary figure legend (“The fraction significantly drops at >10 bp, reflecting limitations of read-based phasing.”) to explicitly state that fact.

4. How often are there more than 2 variants within 10bp? This should be explicitly noted. If the phasing results are sufficient, it would be nice to see this over longer distances.

Our analysis allows us to estimate that out of all the MNVs within a single codon ($n=31,575$), 0.72 % ($= 228/31,575$) are also observed as MNVs consisting of more than 2 variants.

The focus in this paper is on MNVs predicted to have different functional consequences compared to what would be predicted from their constituent SNVs. As such, while we do have a list of MNVs that consist of three SNVs within a single codon ($n=228$), we have not performed an exhaustive inference of MNVs of size >2 across the rest of the genome; performing this calculation would be very computationally expensive, which we believe would not be justified in terms of increased biological understanding.

5. It is noted that only ~0.005% of all possible MNVs were detected in this study. Presumably if the entire population of the world were sequenced, we still would not get every possible MNV across the genome.

This is a great point, and indeed we confidently expect that to be true. We have added a figure plotting the number of samples against the number of MNVs discovered (by downsampling of the gnomAD data set), both in real and log scale (Supplementary Fig. 11 C and D, included below).

If we make a strong assumption that the log-linearity we observe in the current sample size holds true even with orders of magnitude higher sample size, we expect the percentage of all possible MNVs that we could observe from sequencing the entire human population would be roughly 1.7%, as shown in the figure below. We have included this in the legend of Supplementary Fig. 11 C and D, but given the considerable uncertainty associated with the linear assumption above, we have not emphasized this figure in the manuscript.

Figure S11. percentage of MNVs observed

c, d, Number of MNV that are observed, as a function of down-sampled population size (median of 3 different random seeds. Error bars, denoting the maximum and minimum of 3 seeds, were constantly smaller than the dot size), in linear (**c**) and log (**d**) scale. If we make a strong assumption that the log-linearity we observe in the current sample size holds true even with sample sizes an order of magnitude larger, we expect the percentage of all possible MNVs that we observe when we sequence the entire population ($n=7.7 \times 10^9$) would be roughly 1.6%

Projection of the human population size (included for the interests of the reviewer, but not added to the paper):

6. Instead of just considering the functional impact of MNVs within one codon, it would be interesting to know how much of an impact all MNVs produce regardless of whether they are within one codon or across multiple codons. This is a more difficult computational task, but the real question of altered function is of a protein as a whole, not just an individual codon.

We have added a table that describes the number of MNVs in coding region that span across two codons, and showed the breakdown of functional consequence of individual SNVs, as well as the total number of such MNVs (Supplementary Table 5). This is now stated in the “Analysis of functional impact in coding region” section in the methods, as “(See Supplementary table 5 for the number of MNVs that spans across two codons)”.

However, in those pairs of SNVs spanning two codons, functional inference can be performed without explicitly defining MNVs, since these variants act independently in the context of protein translation. Therefore, although we agree these are biologically interesting (e.g. in some cases there may be biochemical interactions between pairs of adjacent missense variants on the same haplotype), there is no clear way to determine which of these variants is most likely to have such an effect. As such we do not perform any further functional characterization of this class of MNV in this manuscript.

In response to this reviewer question, as well as an independent reviewer request for the gnomAD flagship paper that is being reviewed in parallel, we have considered another class of variant pairs whose combined interpretation can be highly different from either of the individual component variants: insertion/deletion pairs that result in frame restoration (e.g. 4bp deletion + 7bp insertion, resulting in 3bp = 1 amino acid insertion). We have made the list of such variant pairs available up to 30 bp distance (considering the limitation of read based phasing), showed that such indel pairs that rescue HC LoF variants are enriched in constrained genes, and suggested they mainly originate from one-step events. The results are summarized in Supplementary Fig. 3, and are mentioned in the “**Functional Impact of MNVs**” section of the results. We have deleted the description “Although theoretically a combination of insertions and

deletions of different lengths could also change the individual consequence of the variants (for example, an insertion of length 4 followed by a nearby deletion of length 1 results in an insertion of length 3, which restores the codon reading frame), we focused on SNV combinations and did not try to identify such class of variants in this work” that was originally in the methods section from our manuscript.

Table S5. Number of MNVs spanning two codons

a, comparison of numbers of pairs of SNVs falling within a single codon and spanning two codons in gnomAD exome data. **b**, breakdown of functional consequence of SNV1 and SNV2, each in different codons. “Others” includes start lost, stop lost and stop retained variants.

a.

distance \ codon	within codon	spanning two codons	total
d=1	25287	14070	39357
d=2	6288	9359	15647
total	31575	23429	NA

b.

SNV1 (codon1)	SNV2 (codon2)	number
synonymous	synonymous	709
synonymous	missense	7464
synonymous	nonsense	213
missense	synonymous	7789
missense	missense	6442
missense	nonsense	243
nonsense	synonymous	234
nonsense	missense	255
nonsense	nonsense	15
		others: 65
		total: 23429

Figure S3. Properties of frame-restoring indel pairs

a, The number of indel pairs (orange = all, blue = phased) is shown as a function of distance between the indels. We set the threshold distance to be 30bp as there are relatively few indel

pairs past this distance. **b**, The distribution of the distance between indel pairs resulting in frame restoration (exome only, same for c~h). **c**, The distribution of the resulting insertion or deletion length for frame-restoring indel pairs. **d**, The number of frame-restoring indel pairs per gene, and the list of genes with more than six such variants. **e-f**, The allele count distribution of frame-restoring indels (**e**) and the distribution of allele counts divided by the maximum allele count of constituent SNVs (**f**). The value is exactly 1 for 81.5% of overall frame-restoring indel pairs, suggesting that the majority of such indel events are likely the result of one-step mutational event. **g-h**, The mean LOEUF (constraint) score (**g**) and the fraction of LoF-constrained genes for frame-restoring indel pairs (**h**), per combination of LOFTEE filters of the constituent indels.

Response to reviewer 2:

1. The authors made the following statement, "the most frequent MNV pattern is CA->TG substitutions, which are likely to occur as a combination of an A->G transition, followed by a high mutation rate C->T CpG transition." If this two-step process indeed occurs in this particular order, they should be able to find samples with higher frequency of CG (A->G occurring first) but not TA (assuming not everyone has the second mutation given a large sample size) at the CA->TG sites. Is this indeed the case?

This is a great point, and indeed this is very clearly the case. We have added the relevant information in supplementary Fig 4A and also cited this in the main text.

Figure S4. Allele frequency difference of the component SNVs in each MNV pattern, number of adjacent MNVs, and comparison with simulation using a simple probabilistic model

a, Fraction of MNV in which allele count of only one of two constituent SNVs is larger than that of MNV (defined as "SNV excess"), indicating the non-uniform frequency of each step of two-step mutational processes. AC1 always corresponds to allele count of W->Y in WX->YZ. CT->TG in this figure is equivalent to AG->CA in other places (reverse complement was taken in this figure for the consistency with other MNV patterns that introduce CpG intermediate).

2. "To investigate the extent of pol-zeta signature, we calculated the number of MNVs in which the gnomAD allele counts of the constitutive single nucleotide variants are equal (following previous methodology2), and observed that these "one-step" MNVs are significantly enriched in MNV patterns matching the pol-zeta signature..." Why not show that all samples either have WT-WT or ALT-ALT but not WT-ALT (WT-ALT would indicate a two-step process)?

Supplementary Fig. 15 contains this information: we observe that for almost all of the one-step MNVs (except for CG->TA, for which the process is likely dominated by two CpG transitions happening in close succession), the number of WT-ALT is less than 10% of that of ALT-ALT. However, we agree that this figure is not intuitive. We have added a sentence and another supplementary figure (S15a) to show that (as suggested by the reviewer) WT-ALT is very rare for one-step MNVs.

Also, to clarify, the presence of WT-ALT haplotypes in the population does indicate that there was an SNV event that did not result in an MNV at that position, but it is worth noting that this does not necessarily indicate a two-step process, as theoretically this outcome could result from an event in which a one-step MNV event is followed by an SNV that reverts one base to the intermediate stage in a subset of haplotypes in the population, or a recombination event between the two adjacent variants, both of which are extremely unlikely scenarios. However, our data supports the notion that $AC1==AC2$ (i.e. allele counts being equal for the two constituent SNVs) is generally a highly reliable indicator of a one-step MNV event.

Discussion of these issues can be found in the **Defining one-step MNVs and MNVs in repetitive contexts** section of the methods, but we have also added a sentence ("also described in Methods") to the Results to clarify this for readers.

Figure S15. Allele count of individual SNVs for one-step MNV

a, Distribution of allele count of one-step MNV divided by that of constituent SNVs.

3. Finding MNVs in repetitive sequence content is going to be problematic because misalignment can definitely contribute to this. How many of the total MNVs reside within repetitive regions (about 10%?).

As we have shown in Figure 3d and Supplementary Fig. 17 of the original manuscript, this proportion depends heavily on the MNV pattern, with the highest proportion being slightly below 60% for AA->TT MNVs. For most MNV classes, it is less than 20%, and for all MNV patterns combined, it is slightly less than 10% (9.25%, as stated in the original main text “Genome-wide mutational mechanisms of MNVs” section), as you suggested.

We completely agree that there is an increased risk of misalignment errors in highly repetitive regions. Indeed, when we calculated the fraction of MNVs falling in extremely low-complexity regions (LCRs), we found that a substantial fraction of overall MNVs in non-coding regions (5% for adjacent MNVs) were in such LCRs. As we might expect, this fraction was especially high for what we define as “MNVs in repetitive regions”, where it reaches 27%.

We agree with the reviewer that variant calls in these repetitive regions are highly enriched for errors. As a result, we decided to filter out MNVs in LCRs entirely from our analysis, and re-performed all analyses in the paper using this filtered data set. To be consistent with the analysis in exome dataset, we also decided to apply individual variant filtering with genotype quality criteria (adj ; GQ >= 20, DP >= 10, and allele balance > 0.2 for heterozygous genotypes).

While this resulted in (typically very small, except for figure 3d) changes to every single number, table and figure in the manuscript, our results, discussion and the major conclusions of the paper remain the same. The changes in the numbers before and after LCR filtering are summarized in the figure below, as well as in Supplementary Table 7 (e.g. After applying LCR filtering and re-defining the repetitive context that are likely to cause polymerase slippage, overall % of MNVs in repetitive regions are 3.15%, and that for AA->TT 30.2%), and a new column in Supplementary Table 2. The fraction of variant filtered out in each filtering step is summarized in Supplementary Fig. 12, and the final fraction of MNVs falling in repetitive contexts are summarized in Supplementary Fig 18 (Note, those MNVs fall in junctions of local repetitive contexts, but are not classified as LCR). Variants in LCRs are flagged but not removed from the release file, so the users who are interested in such variants can still investigate them.

We have described these filtering steps in the **MNV filtering** and **Defining one-step MNVs and MNVs in repetitive contexts** sections in the methods, but of course this filter is not perfect, and there will remain some degree of sequencing artifacts in the MNVs we classify as likely polymerase slippage errors. To clarify this for readers we have added a sentence to the mutational mechanisms section of the results that reads “These findings come with the caveat that variants in repetitive regions will have higher error rates due to slippage and misalignment errors (Supplementary Fig. 9), but we have reduced this risk by applying random forest filtering for individual sites, as well as removing all the variants in low-complexity regions from our analysis (see Methods).”

a.

b.

c.

Figure. Comparison of the effect of LCR / adj filtering on figure 3 b, c and d of the main text (For the interest of reviewers, not in the final manuscript)

a, The number of MNVs, before and after LCR/adj filtering (corresponding to figure 3b ; The blue colored bars, labeled with “post-filtering” corresponds with figure 3b after in the final manuscript, and the orange bars with figure 3b in the old manuscript). The effect is very small and the order itself does not change.

b, The number of one-step MNVs, before and after LCR/adj filtering. The left panel is based on the order and color of the previous figure, and the right panel is in the order and color of the updated figure. After applying the filter, one-step MNVs in AC->GT increases, but the essential information (those involving transition in CpG has the lowest, and pol-zeta signature has high fraction of one-step MNV) remains the same.

c, The number of MNVs in repetitive contexts that are likely to cause polymerase slippage, before and after LCR/adj filtering and updating the definition of repetitive contexts. Left panel is based on the order and the color of the previous figure, and the right panel is in the order and color of the updated figure. Some of the top results before LCR/adj filtering (e.g. AA->CC) disappears, and thus are not defined as repetitive contexts in the new definition, but AA->TT, AT->TA, and TA->AT remains to be in the top and are still defined as repetitive contexts.

Table S7. The number and % of SNV pairs that are filtered out by LCR / adj filtering

Category	Subcategory	number of SNV pairs pre-LCR and adj filtering	number of SNV pairs in LCR / fails adj	% filtered because of in LCR / fails adj
Exome	within codon, 2SNVs	31667	92	0.29
	within codon, 3SNVs	229	1	0.44
Genome, adjacent MNVs	all	1287642	64636	5.02
	one-step MNVs	503189	7695	1.53
	In repetitive context	48885	10381	21.2
	Most common (CA->TG)	275237	5166	1.88
	Least common (TA->GC)	1025	37	0.98
Genome, all MNVs of distance <11	all	6261326	748107	12.0
	one-step MNVs	1031038	25165	2.44

	In repetitive context	502925	199069	39.6
--	-----------------------	--------	--------	------

Table. showing the fraction of variant that are filtered by LCR/adj, in the original definition of MNVs in repetitive contexts (For reviewer's interest, not in the final manuscript. We updated the definition of MNVs in repetitive contexts after filtering LCR/adj, for the final version of manuscript.)

Category	Subcategory	number of SNV pairs pre-LCR and adj filtering	number of SNV pairs in LCR / fails adj	% filtered because of in LCR / fails adj
Genome, adjacent MNVs	In repetitive context (original definition)	119075	32105	27.0
Genome, all MNVs of distance <11	In repetitive context (original definition)	704463	405675	57.6

Table S2. Description of the functional annotations used in the research

The interval length, percentage of the whole genome, mean (across sites) of the median coverage (across individuals) in gnomAD, the mean (across sites) methylation level of CpG sites, and the percentage of regions that falls in LCR, are annotated as separate columns.

Category	Interval length	% of genome	Coverage	Methylation level	% in LCR (filtered)
TSS	55509841	0.019603	30.6	0.194	1.306
5' UTR	19902177	0.007028	30.1	0.350	0.880
Promoter	94046133	0.033212	30.4	0.437	1.793
Enhancer	126128190	0.044541	31.1	0.448	1.407
TFBS	380345514	0.134316	31.2	0.502	1.520
H3K4me3	397134049	0.140244	31.3	0.532	1.664
DHS	492285933	0.173846	31.3	0.543	0.844
H3K9ac	376617506	0.132999	31.2	0.546	1.512
Coding	61730033	0.021799	31.2	0.584	0.721

H3K27ac	772035773	0.272638	30.9	0.664	2.002
H3K4me1	1242793221	0.438882	31.1	0.717	1.590
3' UTR	40715655	0.014378	30.7	0.720	1.243
Intron	1123766231	0.396848	30.6	0.797	2.221
Transcribed	1022244971	0.360997	30.4	0.866	0.880

a.

b.

c.

Figure S12. Fraction of variant pairs that are filtered out by random forest (a) / by regional filter (LCR= low complexity region) (b) / by failing the adjusted criteria (c), up to 10 bp

Row denotes the distance between two SNVs of variant pairs, and column denotes the MNV pattern (and reverse complement). The fraction is represented as color. The denominator is the number before filtering in (a), the number after applying random forest filtering in (b), and the number after applying random forest and LCR filtering in (c).

Figure S18. Fraction of MNV that are in repetitive contexts, up to 10 bp

Row denotes the distance between two SNVs of variant pairs, and column denotes the MNV pattern (and reverse complement). The fraction is represented as color.

4, The axes for figure 3b-d are slightly confusing when they switch from number to fraction in panel c and d. Best to stick with numbers for consistency.

We agree that this is confusing, and have explored multiple options to clarify this for the reader. Our challenge in this figure is showing the number in all cases actually obscures some key aspects of the data because the number of CpG-driven events is enormously higher than other classes. Therefore we have continued to show fraction in panels c and d, but changed the display in these panels to dot plots, so that the readers can more intuitively distinguish the panels showing number (which use histograms) versus fraction (which use dots). We applied this criteria to Figure 2 and Supplementary Fig. 2 as well (except for Supplementary Fig 2b, where dot plots would be too dense and confusing). We have also noted “Error bars represent standard error of the mean (often smaller than the dot size)” to clarify the error estimation. We believe this adds visual consistency to the figures and hopefully reduces confusion for readers.

Figure 3. Mutational origins of MNVs

c, The fraction of one-step MNVs per substitution pattern. Error bars represent standard error of the mean (often smaller than the dot size)[WQ1]. **d,** The fraction of MNVs that are in repetitive contexts, and bits representation³⁸ of sequence contexts. Error bars represent standard error of the mean. Colors in the bars in **(b)** to **(d)** represents the predicted major mechanism of MNVs for each substitution pattern.

5. How many of the 1,996,125 MNVs with constituent variants fall within 2 bp of each other are singletons (seen only in one sample)? What is the "allele frequency" distribution of these MNVs? Similarly, of the 6,261,326 MNVs with constituent variants fall within 10 bp of each other, how many are singletons and what is the "allele frequency" distribution? How many MNVs on average are found in one exome? In one genome? Are their ethnic differences? This information will help the readers understand the scope of this variant class.

These are excellent suggestions. The average number of MNV within codon per individual can already be found at **Functional impact of MNVs** section of the results (“There was an average of 55.2 variants with altered functional interpretation (including 0.062 gained and 4.42 rescued nonsense) due to MNVs per individual”).

The overall allele frequency distribution can be found at Supplementary Fig. 16. However we do agree that the figure is not easy to follow, and more information could be provided. We have thus added the following information to Supplementary Fig 16:

- a panel showing the allele count distribution for different MNV patterns characterized by different potential mechanisms;
- another panel showing the sample size and the number of MNVs discovered for each population;
- another panel showing the proportion of singleton MNVs, as well as the proportion of MNVs in which homozygous carrier(s) are observed, within each population.

We have also added Supplementary Table 6, which summarizes the number of MNV in each consequence category, for each population.

These results fit well with our understanding from standard SNV/indel analysis, showing features such as higher proportion of homozygous carriers in bottlenecked populations (Finns).

We will also release the ethnicity distribution of each MNV in the public gnomAD browser before publication of this manuscript.

b.

c.

Figure S16. Distribution of allele counts of MNVs, and comparison across populations
b, The allele count distribution for different adjacent MNV patterns characterized by different potential mechanisms. Proportion of singleton for Ti at CpG (0.471) is nearly identical to that of repeat (0.472). **c**, The sample size and the number of MNV discovered for each population. **d**, **e**, The proportion of singleton MNVs (**d**) and the proportion of MNVs in which homozygous carrier(s) are observed (**e**) within each population. nfe=European (non-Finnish), amr=Latino, sas=South Asian, fin=European (Finnish), eas=East Asian, afr=African/African American, asj=Ashkenazi Jewish, oth=Others, ordered by the population size.

Table S6. Number of MNVs in each consequence category, for each population

categ\pop	nfe	amr	sas	fin	eas	afr	asj	oth	all
gained nonsense	203	69	86	20	38	65	18	35	407
gained missense	27	16	21	4	8	6	2	4	73
changed missense	6969	3199	3214	950	2100	2559	835	1587	14103
partially changed missense	1103	515	504	149	328	409	128	255	2194
lost missense	80	29	39	8	25	29	11	19	156
Unchanged	6408	3078	3113	879	1913	2519	689	1445	12819
Rescued nonsense	950	407	457	114	283	323	92	177	1821
total	15740	7313	7434	2124	4695	5910	1775	3522	31573*

*2 variants, one with gained stop loss, the other with rescued stop loss, are excluded in this table

6. I am surprised that the phasing sensitivity is only 85% for adjacent heterozygous variant pairs. Since exomes and genomes are typically with >30X coverage, there must be enough high quality reads that cover adjacent variants. An explanation of the low sensitivity is needed.

Yes, we were also initially surprised by these numbers and spent some time exploring the origins of the reduced sensitivity. As mentioned in the main text, we believe this reflects the very stringent phasing criteria employed by GATK. Specifically, as shown in supplementary table 1d and stated in the **Analysis of phasing sensitivity** section of the methods, we found the fraction of unphased variant pairs becomes low not only when the read depth is low, but also when it is too high. This appears to be because when read depth is high, the probability of a base error that is inconsistent with the haplotype phase increases. This error mode is also mentioned as the basis of the idea of maximum read depth filter in the reference 59: Li, H. Toward better understanding of artifacts in variant calling from high-coverage samples. *Bioinformatics* **30**, 2843–2851 (2014). It is clear that better error modeling within GATK could thus improve phasing sensitivity, but such work falls outside the scope of this paper, and the application of such new methods would require complete re-calling of all gnomAD samples.

Other edits made to the manuscript:

In addition to the revisions described above, we have also made some additional minor changes to the manuscript:

- We now include the MNVs in hemizygotes in the analysis. This resulted in a slight increase in the number of MNVs we call and analyze in the exome dataset (from 31510 to 31667 pre-LCR filtering, and to 31575 after LCR filtering).
- We changed the lower y-axis limit of figure 3a to be 1 instead of 10^2 , to avoid overinterpretation.
- We changed the upper y-axis limit of figure 3d to be 1 instead of 0.6, for the consistency of the scale with figure 3c.
- In figure 3 c and d, we changed the number of top and bottom MNV patterns to show, to make it consistent across the paper. Specifically, we show top 6 and bottom 4 MNV patterns. We also updated the legend accordingly, and made it consistent with the MNV pattern classification in figure 4a.
- We corrected the number of MNVs consisting of 3 SNVs to 229 (228 after LCR filtering).
- In the Fig 4d and section **Functional enrichment** in the method, we included the pattern TA->CG as a transversion signal by error, where the correct pattern is TA->GC. We have corrected this in the method section and re-generated the figure 4d, which resulted in a very minor change in the figure (minor reduction of the estimated fraction of transversion origin).
- There was a small bias in the quantification of repetitiveness in the case of $d > 1$, where the number of dinucleotide repeats were counted for the 2 bp context including the upstream constituent SNV of the MNV, but not for the downstream constituent SNV. We have corrected this and added a line in the method section that clearly describes our quantification of repetitiveness. This resulted in a minor change in the Supplementary

Fig. 17 and 18, without major change in the biological interpretation. We also changed the threshold for the quantification of repetitiveness, after applying the LCR filtering.

- We have made a minor change in the mathematical notation in the supplementary methods section. Specifically, what we previously wrote as x/y is now denoted as $\frac{x}{y}$, to better represent that it is x divided by y .
- When explaining the estimation of MNV mutation rate per generation, we mis-annotated the digit of the number of reference base pairs. We fixed this and changed the “Given that there are roughly 1.68×10^9 GA pairs and 1.20×10^9 GC pairs in the reference human genome, we estimate there are on average 0.22 GA->TT and 0.40 GC->AA mutations per generation (Supplementary File 3).” in the main text to “Given that there are roughly 1.66×10^8 GA pairs and 1.20×10^8 GC pairs in the reference human genome, we estimate there are on average 0.026 GA->TT and 0.049 GC->AA mutations per generation (Supplementary File 3). ”.
(Not only the digit but the number itself slightly changes because of the LCR filtering we applied in the revision, but this does not affect any major conclusions)

These changes have no material consequences for the major results discussed in the manuscript.

Reviewers' Comments:

Reviewer #1:

Remarks to the Author:

The authors have made all of the necessary changes to the manuscript and the manuscript is now ready for publication.

Reviewer #2:

Remarks to the Author:

The authors have done a thorough and thoughtful job in responding to the reviewers' comments. The revised paper is more accessible to the readers and provides a definitive look at this class of variants.